# THROUGH THE STEALTH LENS: ATTENTION-AWARE DEFENSES AGAINST POISONING IN RAG

## ABSTRACT

Retrieval-augmented generation (RAG) systems are vulnerable to attacks that inject poisoned passages into the retrieved context, even at low corruption rates. We show that existing attacks are not designed to be stealthy, allowing reliable detection and mitigation. We formalize a distinguishability-based security game to quantify stealth for such attacks. If a few poisoned passages control the response, they must bias the inference process more than the benign ones, inherently compromising stealth. This motivates analyzing intermediate signals of LLMs, such as attention weights, to approximate the influence of different passages on the response. Leveraging attention weights, we introduce the **Normalized Passage Attention Score** (NPAS) and a lightweight **Attention-Variance Filter** (AV Filter) that flags anomalous passages. Our method improves robustness, yielding up to $\sim$ **20%** higher accuracy than baseline defenses. We also develop adaptive attacks that attempt to conceal such anomalies, achieving up to **35%** success rate and underscoring the challenges of achieving true stealth in poisoning RAG systems.

## 1 INTRODUCTION

Large Language Models (LLMs) have revolutionized various applications with their remarkable generative abilities. However, their reliance on internal knowledge can lead to inaccuracies due to outdated information or hallucinations (Achiam et al., 2023; Brown et al., 2020; Ji et al., 2023). RAG (Guu et al., 2020; Lewis et al., 2020) has emerged as a leading technique to address these limitations by integrating LLMs with external (non-parametric) knowledge retrieved from databases (Borgeaud et al., 2022; Karpukhin et al., 2020). It retrieves a set of relevant passages from a knowledge database, denoted as the *retrieved set*, and incorporates them into the model's input. This powerful approach underpins critical real-world systems, including Google Search with AI overviews (Google, 2024), WikiChat (Semnani et al., 2023), Bing Search (Microsoft, 2024), Perplexity AI (Perplexity AI, 2024), and LLM agents (Liu, 2022; LangChain, 2024; Shinn et al., 2023; Yao et al., 2023).

The reliance of RAG systems on the retrieved set, however, introduces a significant new security vulnerability: the knowledge database becomes an additional attack surface. Malicious actors can inject harmful content, for example, by manipulating Wikipedia pages, spreading fake news on social media, or hosting malicious websites, to corrupt the information retrieved by the RAG system (Carlini et al., 2024). Consequently, the retrieval of malicious passages by a RAG system, followed by their incorporation into response generation, constitutes a *retrieval corruption attack* (Xiang et al., 2024). Recent instances, such as the PoisonedRAG attack (Zou et al., 2024), demonstrate easily exploitable vulnerabilities: the attacker simply prompts GPT-4 to create the malicious context and inject it into the retrieved set, successfully manipulating the answer by corrupting only a small fraction of the retrieved set (e.g., one or two out of ten) (Greshake et al., 2023; Zou et al., 2024; Xiang et al., 2024).

Although existing attacks on RAG systems often achieve high success with low corruption rates, they are typically not designed with stealth in mind, leaving them susceptible to detection and mitigation. Ideally, a robust aggregation mechanism would identify inconsistencies between the LLM's output and the dominant (benign) signal in the retrieved set. A significant divergence suggests undue influence from a small, potentially malicious subset of passages. Crucially, to override the benign context, adversarial passages must disproportionately influence the LLM's response. This may necessitate detectable differences from benign passages, leaving behind a malicious trace. The presence of such malicious traces becomes more likely when the adversary cannot compromise the

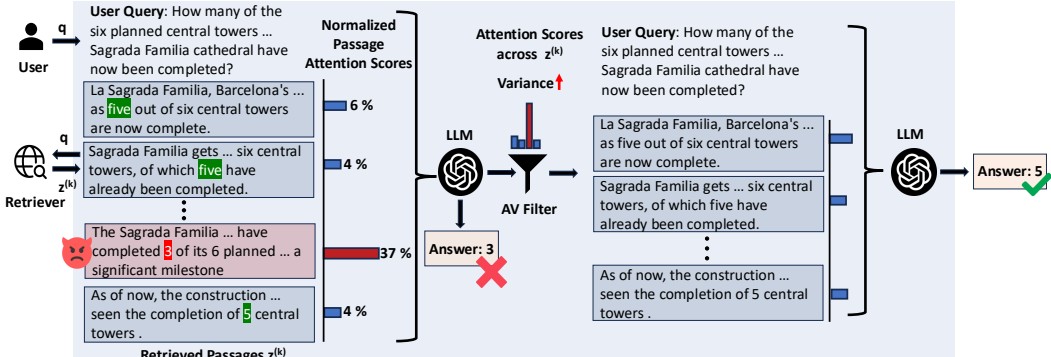

Figure 1: **AV Filter Overview.** The retriever returns passages $z^{(k)}$, one of which is poisoned and disproportionately influences the response, increasing variance in NPAS across passages. AV Filter mitigates this by removing passages with anomalously high attention scores, indicative of poisoning.

majority of the retrieved set, a particularly challenging task when retrieval is performed over large, diverse corpora like Google Search or Wikipedia (Xiang et al., 2024; Zou et al., 2024; Greshake et al., 2023), or when the retriever is designed to be robust. However, existing attacks largely overlook stealth, relying on weak signals such as perplexity (Jain et al., 2023; Alon & Kamfonas, 2023; Gonen et al., 2022). This raises a fundamental question: *Are existing attacks truly stealthy? If not, can they be detected and mitigated, and how can we develop more sophisticated strategies to enhance their stealth?* We challenge the notion of effortless stealth and define it through a distinguishability security game. We introduce the **Normalized Passage Attention Score (NPAS)**, a metric that aggregates the attention weights assigned to tokens in each passage from the model's response. We demonstrate that existing low-effort attacks leave detectable traces, as adversarial passages attract disproportionately high attention, typically due to phrases containing or strongly implying the adversarial answer.

Leveraging the skewed distribution of normalized passage attention scores across the retrieved set, we propose the **Attention-Variance Filter (AV Filter)**—an outlier filtering algorithm that removes passages corresponding to extreme outliers in normalized passage attention scores (See Figure 1 for an overview). The AV Filter effectively distinguishes malicious passages from benign ones, enabling robust defenses by filtering out potentially malicious passages. To rigorously explore the limits of this defense, we extend jailbreak methodologies to create adaptive attacks that optimize for obscuring attention-based traces and evading the AV Filter, marking progress toward stealthier attacks. Our findings highlight the ongoing arms race between attacks and robust RAG systems by formalizing a security game, demonstrating effective mitigation of existing low-stealth attacks, and revealing the challenges in improving stealth through adaptive attacks. We summarize our contributions as follows:

- We formalize stealth in RAG attacks via a distinguishability-based security game.
- We introduce the **Normalized Passage Attention Score** to quantify passage-response dependencies and show that its distribution becomes skewed under corruption.
- Leveraging this score, we design a successful defender for the security game and propose the **Attention-Variance Filter** to identify and remove potentially malicious passages.
- We design stealthier adaptive attacks by leveraging optimization techniques for jailbreaking to evade the Attention-Variance Filter, highlighting the trade-offs in improving stealth.

## 2 BACKGROUND AND RELATED WORK

**Notations and Definitions.**

1. $\mathcal{Q}$: space of queries; $\mathcal{S}$: space of responses. A query $q \in \mathcal{Q}$, a valid response $s \in \mathcal{S}$, and an adversary's target response $s' \in \mathcal{S}$ with $s \neq s'$.

2. $\mathcal{Z}$: space of knowledge databases. A database $z \in \mathcal{Z}$ is a collection of passages $z = \{z_1, z_2, \ldots, z_n\}$, where each $z_i$ is a passage.

3. $z^{(k)} \subset z$: a subset of $k$ retrieved passages. $\mathcal{Z}^{(k)}$: the space of all such subsets.

4. $\Theta$: space of RAG architectures. An architecture $\theta \in \Theta$ is defined as $\theta = (\text{Ret}_\theta, \text{Gen}_\theta, \text{LLM}_\theta)$, consisting of a retriever, generator, and language model.

**Retrieval-Augmented Generation.** A RAG pipeline comprises four key components: a knowledge database, a retrieval function, a generation function, and an LLM. The knowledge database consists of a collection of passages sourced from diverse repositories such as Google Search or Wikipedia.

*Step I. Knowledge Retrieval:* The retriever selects the top-$k$ passages relevant to $q$. Formally, $\text{Ret}_\theta : \mathcal{Q} \times \mathcal{Z} \to \mathcal{Z}^{(k)}$ denotes the retrieval function that returns the top-$k$ passages.

*Step II. Generation*: The generation function utilizes the retrieved set and the LLM, often guided by an instructional prompt $\mathcal{I}$, to produce the final response. Formally, $\text{Gen}_\theta : \mathcal{Q} \times \mathcal{Z}^{(k)} \to \mathcal{S}$ denotes the generation function that outputs the response $s$.

> A Retrieval-Augmented Generation (RAG) system can be formally defined as the function $f_{\text{RAG}} : \mathcal{Q} \times \mathcal{Z} \times \Theta \to \mathcal{S}$, where $f_{\text{RAG}}(q, z, \theta) = \text{Gen}_\theta(q, \text{Ret}_\theta(q, z)) = s$.

In a standard RAG pipeline (Lewis et al., 2020), the retriever assigns relevance scores to passages independently and selects the top-$k$ passages based on these scores. The retriever's output is:

$$\text{Ret}_\theta(q, z) = z^{(k)} = \{z_{i_1}, z_{i_2}, \ldots, z_{i_k}\}$$

Next, the generation function processes a concatenated sequence consisting of the instructional prompt, the retrieved passages, and the query, to produce a response. This is formulated as:

$$\text{Gen}_\theta\left(q, z^{(k)}\right) = \text{LLM}_\theta\left(\text{Concat}(\mathcal{I}, z^{(k)}, q)\right) = \text{LLM}_\theta(\mathcal{I} \oplus z_{i_1} \oplus z_{i_2} \oplus \cdots \oplus z_{i_k} \oplus q),$$

where $\oplus$ denotes the concatenation of text sequences.

**Vulnerabilities in RAG Systems.** An adversary targeting a specific response $s'$ can craft a set of adversarial passages $z_{\text{adv}}$ by simultaneously maximizing the following two objectives:

$$\Pr_\theta\left[z_{\text{adv}} \subset \text{Ret}_\theta\left(q, z \cup z_{\text{adv}}\right)\right] \quad \text{and} \quad \Pr_\theta\left[\text{LLM}_\theta\left(\text{Concat}(\mathcal{I}, z^{(k)}, q)\right) = s' \,\middle|\, z_{\text{adv}} \subset z^{(k)}\right]$$

These correspond to attacks on *Step I* and *Step II*, respectively. Existing RAG systems are highly brittle to poisoning, and even minimal corruption—for example, altering a single passage among ten retrieved—can successfully manipulate LLM responses. Given the open challenge of building perfectly robust retrievers (Fayyaz et al., 2025; Lin, 2024; Li et al., 2021), enhancing robustness at the generation stage (*Step II*) becomes critical. This allows tolerance to limited corruption and enables reliable integration with reasonably robust retrieval methods such as Google Search, yielding an end-to-end robust RAG pipeline. **This work focuses on strengthening the robustness of the generation stage.** We argue that a notion of stealth improves robustness by allowing generation to withstand small-scale corruption. Advancing the robustness of retrievers is an orthogonal challenge with broader applications and sensitivity to corpus characteristics; we defer improving and analyzing weak retrievers, such as BM25, for integration into robust end-to-end pipelines to future work.

**Existing Work.** QA models are vulnerable to disinformation attacks (Du et al., 2022; Pan et al., 2021; 2023; Zhong et al., 2023), with recent work highlighting risks specific to RAG pipelines. We categorize attacks into: (i) *content-poisoning* methods that inject incorrect information into retrieved passages (often LLM-generated) to bias the generation towards an adversary-specified answer (e.g., **PoisonedRAG (Poison)** (Zou et al., 2024), **Misinformation Attack (MA)** (Pan et al., 2023), and **RAG Paradox (Paradox)** (Choi et al., 2025)), and (ii) *instruction-poisoning* methods that embed direct prompt within the retrieved passages to elicit incorrect responses (e.g., **Prompt Injection Attack (PIA)** (Greshake et al., 2023)). Although the former may appear more stealthy to human readers, we show that both classes leave comparably detectable traces in the internal representations of LLM when attempting to steer inference toward incorrect outputs. *Our analysis is not restricted to particular attack types, but rather investigates how poisoned passages perturb intermediate LLM representations compared to benign passages, independent of surface-level semantics.*

In response to these threats, several strategies have been explored, including query paraphrasing (Weller et al., 2022), misinformation detection (Hong et al., 2023), vigilant prompting (Pan et al.,

2023), reranking methods (Glass et al., 2022), and perplexity-based filters (Jain et al., 2023; Alon & Kamfonas, 2023; Gonen et al., 2022). However, these methods often suffer from limited efficacy or high false positive rates (Zou et al., 2024). More recently, Certified Robust RAG (Xiang et al., 2024) was introduced, which employs an isolate-then-aggregate strategy to provide empirical accuracy bounds and reduce attack success, and it currently represents the state of the art in mitigating such threats. Further details on the existing work are provided in Appendix A.

## 3 STEALTH ANALYSIS OF ADVERSARIAL ATTACKS IN RAG SYSTEMS

**Threat Model.** We consider an adversary $\mathcal{A}_\epsilon$ with full knowledge of the RAG architecture $\theta$ and knowledge base $z$. The adversary may inject (but not modify or delete) up to $\lfloor \epsilon \cdot k \rfloor$ poisoned passages into $z$. Given a query $q$, target $s'$, and architecture $\theta$, the adversary produces a poisoned set $z_{\text{adv}}^{(\lfloor \epsilon \cdot k \rfloor)} = \mathcal{A}_\epsilon(q, z, s', \theta) = \left\{ z_1, \ldots, z_{\lfloor \epsilon \cdot k \rfloor} \right\}$, constructed so that all poisoned passages are retrieved and collectively induce generation of $s'$. Let $z_{\text{benign}}^{(k)} = \text{Ret}_\theta(q, z)$ and $z_{\text{corrupt}}^{(k)} = \text{Ret}_\theta \left( q, z \cup z_{\text{adv}}^{(\lfloor \epsilon \cdot k \rfloor)} \right)$ denote the top-$k$ retrieved passages from the benign and poisoned knowledge bases, respectively. Then the retrieved sets may differ by at most $\epsilon \cdot k$ passages. The attack is successful if the RAG system generates the target response, i.e., $\text{Gen}_\theta \left( q, z_{\text{corrupt}}^{(k)} \right) = s'$.

We consider a defender $\mathcal{D}$ with full knowledge of the RAG architecture $\theta$ and access to a limited set of benign passages from the knowledge base. This assumption is practical, since defenders are often system developers or model providers with visibility into LLM internals. To test generality, we further extend experiments to settings where the RAG system uses closed-source APIs (e.g., GPT-4).

**Attack Practicality.** We focus on attack scenarios where benign passages in the retrieved set form a clear majority consensus. If poisoned passages become the majority, either because of a weak retriever or a lack of sufficient benign evidence in the knowledge base, generating an accurate response becomes provably impossible. We therefore study the practical setting where $\epsilon < 0.5$, which reflects real-world cases where attackers have limited resources and can inject only a few poisoned passages into the top-$k$ results, such as with web search retrievers. In our experiments, we follow prior work by injecting poisoned passages into a benign retrieved set to form the corrupted set.

**Stealth Attack Distinguishability Game (SADG).** Given a RAG architecture $\theta$ and a knowledge database $z$, we define a game between an arbiter, an adversary $\mathcal{A}_\epsilon$, and a defender $\mathcal{D}$, parameterized by a corruption budget $\epsilon$. The defender does not have access to $z$. The game proceeds as follows:

1. The arbiter samples a query $q$ and constructs the benign retrieved set $z_{\text{benign}}^{(k)} = \text{Ret}_\theta(q, z)$.

2. The adversary $\mathcal{A}_\epsilon$ generates poisoned passages $z^{(\text{adv})}$ under budget $\epsilon$, and the arbiter constructs the corrupted set $z_{\text{corrupt}}^{(k)} = \text{Ret}_\theta \left( q, z \cup z^{(\text{adv})} \right)$.

3. The arbiter sends the query $q$ and the two retrieved sets in random order, as $\left( z_0^{(k)}, z_1^{(k)} \right)$, to the defender. The defender must guess which set is corrupted to win the game.

The defender's advantage is defined as: $\text{Adv}_{\text{SADG}}^{\mathcal{A}_\epsilon, \mathcal{D}}(\theta, z, \epsilon) := \left| \Pr[\text{Defender wins}] - \frac{1}{2} \right|$. Smaller $\epsilon$ implies a tighter corruption budget, making stealth more difficult and increasing Adv. An attack is $\tau$-*stealthy* if, for all probabilistic polynomial-time (PPT) defenders $\mathcal{D}$, the advantage is at most $\tau$. A perfectly stealthy attack corresponds to $\tau = 0$. See Appendix B for details.

**Stealth of Existing Attacks.** While existing attacks may evade detection methods that analyze passages in isolation (e.g., perplexity filtering), their influence is still evident in the model's output, making the generated response itself a valuable signal for detecting corruption. This motivates a shift in perspective: **to analyze retrieved passages in conjunction with the generated response and assess whether any passage disproportionately shapes the output.** If most retrieved passages are expected to be relevant to the query, a strong alignment between the response and only a few passages may indicate adversarial manipulation. We formalize this insight with NPAS, which quantifies the alignment between each passage and the generated response. NPAS enables two defenses: a defender $\mathcal{D}_{\text{AV}}$ that distinguishes between benign and corrupted retrieved sets with a strong advantage in SADG, and the **AV Filter**, which removes potentially poisoned passages to effectively mitigate attacks.

## 4 Stealth Detection and Mitigation via Attention Variance

We build on the fact that, in a successful attack, the generated response is strongly correlated with the malicious passages that shaped it. Ideally, for a retrieved set $z^{(k)}$ and target response $s'$, we should consider the conditional probability $\Pr_{z^{(k)}}\left(\text{Gen}_\theta(q, z^{(k)}) = s' \middle| z_i \in z^{(k)}\right)$ for each passage $z_i$ in a retrieved set to measure its correlation with the response.

**Why analyze attention scores?** In transformer-based LLMs, the information from tokens in the prompt is combined to get an internal representation used for next-token generation. When the predicted token is adversarial, the information needed to generate it must primarily come from poisoned passages. Transformers pass this information between tokens through attention, and across blocks via MLPs and residual connections. The magnitude of information flow through an attention head from another token is proportional to its attention score, as these scores weight the sum of prior token representations to produce the next-block representation of a particular token. The final-block representation of the last token is then used to produce a new token, aggregating dependencies across all tokens. Thus, attention scores computed during inference provide a useful approximation of inter-token dependencies and are widely used for analyzing them (Vig & Belinkov, 2019). When malicious tokens are generated, the internal representation is biased toward poisoned passages, producing skewed attention patterns. We therefore analyze the attention matrices of the LLM to approximate these correlations. This approximation can be further refined using techniques such as attention rollouts (Abnar et al., 2020) or other saliency methods, which we leave for future work.

Analyzing the attention matrix of LLMs in RAG systems has proven useful beyond security considerations, for example, in optimizing KV caches during inference (Zhang et al., 2023; He et al., 2024). $H_2O$ (Zhang et al., 2023) shows that only a small fraction of input tokens, termed *Heavy Hitters* ($H_2$), dominate attention weights when generating a new token. These Heavy Hitters naturally emerge and are strongly correlated with token co-occurrence. Consistent with this, our analysis of compromised RAG systems finds that when malicious influence leads to incorrect responses, the Heavy Hitters are localized within the poisoned passages. Heavy Hitters are often target-response keywords embedded within poisoned passages. These tokens, due to their co-occurrence with the incorrect generated output, receive disproportionately high attention, skewing the overall attention distribution.

Based on this insight, we define the NPAS, which aggregates token-level attention to quantify the proportion of total attention each passage receives from the final response. This score helps identify anomalous passages indicative of adversarial influence. This skewed distribution of attention in poisoned passages is illustrated through examples in Appendix C.1.

**Normalized Passage Attention Score.** Let the input to $\text{LLM}_\theta$ be $\mathcal{X} = \text{Concat}(\mathcal{I}, z^{(k)}, q)$ where $z^{(k)}$ is the retrieved set and $q$ is the query. It generates a response $s' = \{s'_1, s'_2, \ldots, s'_l\}$ of $l$ tokens while computing multi-layer, multi-head attention weights, with each layer producing a separate tensor for each head. We average these weights across all decoder layers and heads to construct a unified attention matrix: $A = \text{Attention}(\text{LLM}_\theta, \mathcal{X}) \in \mathbb{R}^{l \times T}$ where $T$ is the number of input tokens. Each entry $A[i, j]$ denotes the mean attention from the $i$-th output tokens to the $j$-th input token. This averaging yields a stable view of token-level interactions (Peysakhovich & Lerer, 2023).

Each retrieved passage $z_t$ is a finite sequence of tokens, $z_t = \{z_t^{(1)}, z_t^{(2)}, \ldots\}$. The **Passage Attention Score**, $\text{Score}_\alpha(z_t, A)$, is defined as the total attention from all response tokens $s'$ to the top-$\alpha$ most attended tokens in $z_t$, denoted as $\text{Top}_\alpha(z_t)$. This focuses on high-signal Heavy Hitter tokens—often adversarial keywords—within a passage thereby amplifying adversarial cues and reducing noise. We define the **Normalized Passage Attention Score** (NPAS), $\text{NormScore}_\alpha(z_t, z^{(k)}, A)$, by dividing each passage's score by the total score across all $k$ retrieved passages. While normalization preserves ranking, it standardizes attention magnitudes across queries and models, enabling a stable threshold for detecting adversarial passages—unlike instance-specific approaches (Xian et al., 2025). For clarity, we rescale it to a percentage and refer to it simply as a passage's attention score.

$$\text{Score}_\alpha(z_t, A) = \sum_{i=1}^{l} \sum_{x_j \in \text{Top}_\alpha(z_t)} A[i, j] \qquad \text{NormScore}_\alpha(z_t, z^{(k)}, A) = \frac{\text{Score}_\alpha(z_t, A)}{\sum_{i=1}^{k} \text{Score}_\alpha(z_i, A)}$$

We compute the attention score of a passage by summing the top-$\alpha$ most attended tokens within it. For any fixed $\alpha$, this score remains invariant to the passage length, preventing adversaries from gaining an

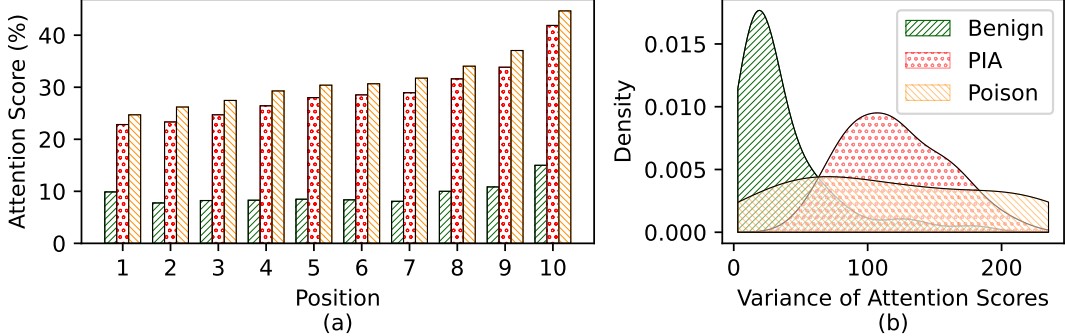

Figure 2: (a) Average attention scores across passage positions in retrieved sets over multiple queries. Benign passages show nearly uniform scores, while a poisoned passage at any position receives disproportionately high attention. (b) Variance of attention scores in benign vs. corrupted sets, showing that a poisoned passage shifts the variance distribution, making it separable from the benign case. Scores are computed on the RQA dataset with Llama 2 for $\alpha = \infty$ and $\epsilon = 0.1$.

---

**Algorithm 1: A**ttention-**V**ariance **Filter** (AV Filter)

---

**Input:** Query $q$, Retrieved set $z^{(k)}$, model $\text{LLM}_\theta$, Corruption fraction $\epsilon$, Variance threshold $\delta$
**Output:** Filtered set $z_{\text{filtered}}$

1   $z^{\text{sorted}} = \text{Sort}(z^{(k)}, \text{LLM}_\theta)$           ▷ Sort passages according to attention scores
2   $z^{\text{filtered}} \leftarrow z^{\text{sorted}}$              ▷ Initialize the set of filtered passages
3   **while** $|z^{\text{filtered}}| > \lfloor (1-\epsilon) \cdot k \rfloor$ **do**
4      $\mathcal{X} \leftarrow \text{Concat}(\mathcal{I}, z^{\text{filtered}}, q)$            ▷ Form the input sequence
5      $A \leftarrow \text{Attention}(\text{LLM}_\theta, \mathcal{X})$       ▷ Compute the attention matrix from $\text{LLM}_\theta$ on $\mathcal{X}$
6      $\text{attn\_scores} \leftarrow \{\text{NormScore}_\alpha(z_t, z^{\text{filtered}}, A) | z_t \in z^{\text{filtered}}\}$     ▷ Compute attention scores
7      $\sigma^2 = \text{Var}(\text{attn\_scores})$          ▷ Compute the variance of attention scores
8      **if** $\sigma^2 \leq \delta$ **then**
9         **break**
10     $z_{\text{max}} = \underset{z_t \in z^{\text{filtered}}}{\text{argmax}} \ \text{NormScore}_\alpha(z_t, z^{\text{filtered}}, A)$
11     $z^{\text{filtered}} \leftarrow z^{\text{filtered}} \setminus z_{\text{max}}$    ▷ Remove the passage with the highest score from the filtered set
12 **return** $z^{\text{filtered}}$

---

advantage through length manipulation. Ideally, $\alpha$ should match the number of Heavy Hitters—tokens in the poisoned passage that align with the target response—which is often proportional to the number of tokens in the target response. We select sufficiently large values of $\alpha$ to ensure coverage of all Heavy Hitters, using $\alpha \in \{5, 10, \infty\}$, where $\infty$ denotes summing over all tokens in the passage. We provide a detailed rationale for selecting the top-$\alpha$ tokens for NPAS in Appendix C.3.

**Discriminating Between Corrupted and Benign Retrievals via Attention.** In benign RAG instances—where retrieved passages are relevant to both query and response—attention distribution over passages is approximately uniform with a slight *recency effect* (Liu et al., 2023a; Guo & Vosoughi, 2024). Corruption skews this attention pattern—Figure 2(a) shows that corrupting a single index significantly elevates its attention scores relative to the benign baseline. This implies that corrupted retrieved sets exhibit a high variance of attention scores across passages. Motivated by this, we propose a defender $\mathcal{D}_{\text{AV}}$ in the SADG game that detects corruption using attention variance. Given a query $q$ and two retrieved sets $\left(z_0^{(k)}, z_1^{(k)}\right)$, the defender computes attention scores for each passage: $\text{attn\_scores}_i = \left\{\text{NormScore}_\alpha(z_t, z_i^{(k)}, A) | z_t \in z_i^{(k)}\right\}$, and calculates their variance $\text{Var}(\text{attn\_scores}_i)$. The defender then outputs:

$$\mathcal{D}_{\text{AV}}(q, z_0^{(k)}, z_1^{(k)}) := \begin{cases} 0, & \text{if } \text{Var}(\text{attn\_scores}_0) > \text{Var}(\text{attn\_scores}_1), \\ 1, & \text{otherwise.} \end{cases}$$

The defender flags the set with a higher attention score variance as corrupted. Figure 2(b) shows that the variance is consistently higher for corrupted sets, enabling reliable detection across attacks.

**Filtering Poisoned Passages from Corrupted Retrievals.** We propose the **Attention-Variance Filter (AV Filter)**, an outlier filtering algorithm that removes potentially corrupted passages exhibiting unusually high attention. Given a query $q$, retrieved set $z^{(k)}$, model $\text{LLM}_\theta$, corruption budget $\epsilon$ and threshold $\delta$, the filter computes the variance of normalized attention scores and iteratively removes the top-scoring passage until the variance drops below $\delta$ or $\epsilon$-fraction of passages are removed.

To address the recency effect, where a few tokens receive slightly higher attention due to their proximity to the next token being generated, we reorder passages by attention score using $\text{Score}(z^{(k)}, \text{LLM}_\theta)$ (Peysakhovich & Lerer, 2023). This sorting reduces positional bias, amplifies anomalous signals, and improves filtering. Algorithm 1 specifies the AV Filter procedure.

**Estimating the Filtering Threshold $\delta$.** The AV Filter's effectiveness hinges on choosing an appropriate threshold $\delta$. We estimate it using the RQA dataset (Kasai et al., 2023) and Llama 2 (Touvron et al., 2023) by computing attention score variances across clean retrieved sets, setting $\delta$ as the mean plus one standard deviation. In selecting $\delta$, we prioritize minimizing false negatives over false positives, since dropping a few benign passages rarely changes the final response when most content is clean. The estimated threshold generalizes effectively to unseen settings.

## 5 EVALUATION

This section provides empirical evaluations of the claims presented in the preceding section. Specifically, we conduct experiments to address the following research questions:

> **RQ1:** Can the defender $\mathcal{D}_{\text{AV}}$ reliably identify corrupted retrievals in existing attacks?
> **RQ2:** How *effective* is the AV Filter at mitigating existing attacks?
> **RQ3:** How *effective* and *efficient* are adaptive attacks at bypassing the AV Filter?

**Experimental Highlights** We summarize the findings related to the research questions:

**RQ1:** $\mathcal{D}_{\text{AV}}$ identifies the corrupted set and wins the security game against existing attacks with high probability. We estimated its probability of winning as the rate of correct identification across settings, achieving an average of $0.78$—highlighting a strong advantage against existing attacks.

**RQ2:** The AV Filter outperforms baseline defenses, achieving up to $23\%$ higher accuracy in benign settings and up to $20\%$ under attack, while maintaining comparable reductions in attack success rates.

**RQ3:** Adaptive attacks bypass the AV Filter, achieving an ASR up to $35\%$—higher than existing attacks—but the AV Filter nevertheless reduces ASR below that of vanilla RAG and empirical upper bounds of baseline defenses. This success requires costly, query-specific optimization ($\sim 10^3 \times$ runtime of baselines) and access to benign passages, unlike prior manual or one-shot LLM attacks.

### 5.1 EXPERIMENTAL SETUP

**Datasets.** We evaluate on four benchmark question-answering datasets: **RealtimeQA (RQA)** (Kasai et al., 2023), **Natural Questions (NQ)** (Kwiatkowski et al., 2019) and **HotpotQA** (Yang et al., 2018) for *short-answer open-domain QA*, and the **RealtimeQA-MC (RQA-MC)** (Kasai et al., 2023) for *multiple-choice open-domain QA*. Each dataset interfaces with a knowledge source: Google Search is used for RQA, RQA-MC, and NQ, while the Wikipedia corpus is used for HotpotQA and also for NQ. We evaluate 100 queries per dataset, following the baseline (Xiang et al., 2024).

**RAG Setup.** We evaluate five LLMs: Llama2-7B-Chat (Touvron et al., 2023), Mistral-7B-Instruct (Chaplot, 2023), Llama-3.1-8B-Instruct (AI, 2024), Deepseek-R1-distill-qwen-7B (Guo et al., 2025), and GPT-4o Achiam et al. (2023). We use the top $k = 10$ retrieved passages. We randomly select Mistral-7B to compute attention scores, while GPT-4o generates the final responses.

**Attacks.** We evaluate three content-poisoning attacks: **Poison** (Zou et al., 2024), **Misinformation Attack (MA)** (Pan et al., 2023) and **Paradox** (Choi et al., 2025), as well as one instruction-poisoning

Table 1: Clean Accuracy (ACC) of defenses, showing that AV Filter preserves RAG utility with a minimal drop from Vanilla, achieving up to 23% higher ACC than other baselines.

| LLM | Mistral-7B | | | Llama2-C | | | GPT-4o | | | Llama-3.1 | | | Deepseek-R1 | | |
|---|---|---|---|---|---|---|---|---|---|---|---|---|---|---|---|
| Defense | RQA-MC | RQA | NQ | RQA-MC | RQA | NQ | RQA-MC | RQA | NQ | RQA-MC | RQA | NQ | RQA-MC | RQA | NQ |
| Vanilla | 81.0 | 72.0 | 62.0 | 79.0 | 61.0 | 59.0 | 66.2 | 69.8 | 61.2 | 44.0 | 71.0 | 64.0 | 37.0 | 56.0 | 54.0 |
| Keyword | 58.0 | 56.0 | 51.0 | 56.0 | 57.0 | 54.0 | **63.2** | **64.2** | **60.4** | **61.0** | 61.0 | 62.0 | 42.0 | 41.0 | 43.0 |
| Decoding | 57.0 | 57.0 | 55.0 | 44.0 | 54.0 | 41.0 | – | – | – | 56.0 | 56.0 | 56.0 | **44.0** | 44.0 | 44.0 |
| **AV Filter**$_{(\alpha=5)}$ | 73.0 | **66.0** | **59.0** | **79.0** | **60.0** | 51.0 | 57.8 | 61.6 | 57.8 | 43.0 | **67.0** | **66.0** | 36.0 | 57.0 | **52.0** |
| **AV Filter**$_{(\alpha=10)}$ | 74.0 | 65.0 | 58.0 | 75.0 | 57.0 | **54.0** | 59.8 | 62.6 | 55.0 | 45.0 | 66.0 | **66.0** | 37.0 | **59.0** | **52.0** |
| **AV Filter**$_{(\alpha=\infty)}$ | **76.0** | 64.0 | 58.0 | 75.0 | 56.0 | **54.0** | 59.6 | 63.0 | 55.8 | 44.0 | **67.0** | 62.0 | 34.0 | 57.0 | **52.0** |

attack, **PIA** (Greshake et al., 2023). Unless otherwise stated, the corruption fraction is set to $\epsilon = 0.1$, with the position of the poisoned passage randomly varied within the retrieved set.

**Defenses.** We evaluate the **AV Filter**, using $\mathsf{NormScore}_\alpha$ for $\alpha \in \{5, 10, \infty\}$. We set $\delta = 26.2$, estimated from benign RQA with Llama2 at $\alpha = \infty$. Baselines include vanilla RAG (**Vanilla**) and Certified Robust RAG (Xiang et al., 2024): **Keyword** and **Decoding**.

**Evaluation Metrics.** For **RQ1**, we measure the success of defender $\mathcal{D}_{\mathrm{AV}}$ in SADG via the **Corruption Identification Rate (CIR)**, the fraction of corrupted sets correctly flagged under successful attacks on vanilla RAG. For **RQ2** and **RQ3**, we report three metrics (percentages): **Clean Accuracy (ACC)**—correct responses without attack; **Robust Accuracy (RACC)**—correct responses under attack; and **Attack Success Rate (ASR)**—responses containing the adversary's target. A response is correct if it contains a valid variation of the ground-truth answer $s$ and excludes the adversary's target $s'$. All results are averaged over 5 random seeds.

We report a representative subset of results with Poison and PIA attacks using Google Search. Expanded results on additional attacks (Appx. D.2), knowledge bases (Appx. D.6), baselines (Appx. D.5), false positive rates (Appx. D.4), hyperparameter analysis (Appx. D.8), ensembling with Certified Robust RAG (Appx. D.7), and other experimental details are provided in Appendix D.

## 5.2 RESULT AND DISCUSSION

**RQ1.** Table 3 in Appendix B reports the estimated probability of $\mathcal{D}_{\mathrm{AV}}$ winning the SADG, measured via CIR, across models, datasets, and varying $\alpha$ under existing attacks. $\mathcal{D}_{AV}$ identifies the corrupted set with high accuracy, achieving an average CIR of **0.78**, demonstrating strong effectiveness. We used successful attack instances against Vanilla RAG in each setting to compute CIR.

**RQ2. Clean Accuracy.** Table 1 presents clean accuracy across models, datasets, and $\alpha$ values. The AV Filter maintains strong clean performance, with an average drop of only **4-6**% across datasets— substantially smaller than other defenses. On RQA-MC, accuracy drops from **61.4**% (Vanilla) to **59.3**% with AV Filter, compared to larger declines for Keyword (**56.0**%) and Decoding (**50.3**%). Similar trends hold for RQA (from **65.9**% to **62.4**%) and NQ (from **60.0**% to **57.76**%).

**Robust Accuracy.** Table 2 reports AV Filter's robust accuracy (RACC) and attack success rate (ASR). On RQA-MC, it achieves **55.7**% RACC, outperforming Vanilla RAG (**44.4**%), Keyword (**53.9**%), and Decoding (**47.1**%). Similar improvements hold for RQA (**59.8**%) and NQ (**53.4**%). AV Filter's RACC closely matches Vanilla's clean accuracy, indicating high precision and minimal benign impact. Appendix D.3 details how often it correctly removes poisoned passages.

**Attack Success Rate.** Table 2 shows that even with a small corruption rate ($\epsilon = 0.1$), Vanilla RAG is highly vulnerable—reaching up to **88.2**% attack success. AV Filter cuts this sharply to an average of **6.6**% on RQA-MC, comparable to Cert. RAG-Keyword (**6.1**%) and Decoding (**7.6**%). Similar trends hold for RQA and NQ, with the average ASR reduced to **6.0**% and **4.8**%, respectively.

Overall, AV Filter mitigates existing attacks while maintaining higher accuracy than Certified RAG. It also requires fewer $\mathrm{LLM}_\theta$ computations, since it avoids evaluating passages individually.

**RQ3.** We adapt GCG (Zou et al., 2023) and AutoDAN (Liu et al., 2023b) by optimizing the poisoned passage with full access to the input. We minimize $\mathcal{L}_1 + \lambda \cdot \mathcal{L}_2$, where $\mathcal{L}_1$ is the cross-entropy loss w.r.t. the target response, and $\mathcal{L}_2$ is the attention variance across passages. Existing attacks yield

Table 2: Robust Accuracy and Attack Success Rate (RACC/ASR) showing that AV Filter effectively mitigates attacks with low ASRs while achieving up to 20% higher RACC than baseline defenses.

| LLM | Dataset
Attack
Defense | RQA-MC | | RQA | | NQ | |
|---|---|---|---|---|---|---|---|
| | | PIA
(racc↑ / asr↓) | Poison
(racc↑ / asr↓) | PIA
(racc↑ / asr↓) | Poison
(racc↑ / asr↓) | PIA
(racc↑ / asr↓) | Poison
(racc↑ / asr↓) |
| **Mistral-7B** | Vanilla | 59.6 / 31.0 | 62.2 / 30.0 | 52.2 / 26.6 | 50.0 / 23.4 | 40.8 / 24.6 | 52.0 / 9.2 |
| | Keyword | 57.0 / 7.00 | 55.0 / **6.00** | 54.0 / 6.00 | 55.0 / **6.00** | 50.0 / **1.00** | 49.0 / **1.00** |
| | Decoding | 55.0 / **5.00** | 54.0 / 13.0 | 55.0 / 5.00 | 54.0 / 13.0 | 55.0 / **1.00** | **56.0 / 1.00** |
| | AV Filter$_{(\alpha=5)}$ | 76.6 / 5.80 | 70.0 / 10.0 | 62.6 / 2.80 | **58.2** / 7.40 | 54.4 / 3.00 | 53.4 / 6.20 |
| | AV Filter$_{(\alpha=10)}$ | **77.2** / 6.00 | 71.6 / 8.20 | 64.8 / 2.80 | 56.8 / 8.40 | **56.2** / 2.80 | 52.8 / 6.60 |
| | AV Filter$_{(\alpha=\infty)}$ | 76.2 / 7.20 | **73.8** / 8.40 | **65.0** / 2.40 | 56.8 / 8.60 | 50.2 / 5.80 | 52.8 / 4.00 |
| **Llama2-C** | Vanilla | 33.4 / 63.0 | 62.8 / 27.6 | 5.80 / 88.2 | 57.4 / 17.2 | 10.6 / 73.2 | **56.8** / 5.80 |
| | Keyword | 54.0 / **6.00** | 53.0 / **5.00** | 53.0 / 6.00 | 53.0 / **5.00** | **52.0** / 2.00 | 51.0 / **2.00** |
| | Decoding | 38.0 / 12.0 | 40.0 / **5.00** | 38.0 / 12.0 | 40.0 / **5.00** | 39.0 / 17.0 | 40.0 / 4.00 |
| | AV Filter$_{(\alpha=5)}$ | 65.6 / 18.4 | 67.8 / 18.4 | **61.8** / 1.60 | 55.4 / 7.00 | 50.6 / 5.20 | 49.8 / 6.20 |
| | AV Filter$_{(\alpha=10)}$ | **70.8** / 12.4 | 69.6 / 13.0 | 60.2 / **1.60** | 54.8 / 8.80 | 51.4 / 4.00 | 51.2 / 6.20 |
| | AV Filter$_{(\alpha=\infty)}$ | 68.8 / 16.8 | **72.0** / 12.6 | 60.2 / 5.00 | **56.8** / 6.60 | 49.4 / 9.20 | 51.8 / 3.60 |
| **Llama-3.1** | Vanilla | 42.0 / 15.0 | 30.6 / 19.2 | 48.4 / 14.0 | 21.0 / 29.4 | 34.6 / 22.4 | 41.0 / 10.8 |
| | Keyword | **61.0** / 7.00 | **58.0** / 6.00 | 61.0 / 7.00 | 57.0 / **6.00** | 60.0 / 3.00 | **58.0 / 2.00** |
| | Decoding | 55.0 / 7.00 | 51.0 / 17.0 | 55.0 / 7.00 | 51.0 / 17.0 | 49.0 / 13.0 | 49.0 / 10.0 |
| | AV Filter$_{(\alpha=5)}$ | 43.0 / **2.60** | 35.8 / 10.6 | 70.2 / **2.60** | 53.8 / 7.20 | **60.8 / 1.00** | 50.2 / 5.00 |
| | AV Filter$_{(\alpha=10)}$ | 44.2 / 2.80 | 36.2 / 7.00 | 67.8 / 3.00 | 53.2 / 6.40 | 57.8 / 2.20 | 50.6 / 5.20 |
| | AV Filter$_{(\alpha=\infty)}$ | 42.2 / 3.20 | 36.4 / **6.00** | 68.2 / 2.80 | **57.4** / 6.20 | 53.8 / 5.20 | 54.4 / 4.20 |
| **Deepseek-R1** | Vanilla | 26.0 / 2.60 | 23.6 / 9.60 | 24.3 / 49.6 | 46.3 / 17.00 | 33.3 / 33.0 | 48.6 / 7.30 |
| | Keyword | 40.0 / 3.00 | 36.0 / 3.00 | 40.0 / 3.00 | 37.0 / 3.00 | **44.0** / 2.00 | 44.0 / 2.00 |
| | Decoding | **42.0 / 1.00** | **42.0 / 1.00** | 42.0 / **1.00** | 42.0 / **1.00** | **44.0 / 1.00** | 43.0 / **0.00** |
| | AV Filter$_{(\alpha=5)}$ | 35.0 / **1.00** | 21.0 / 8.60 | 39.3 / 25.6 | 45.3 / 20.3 | 33.6 / 29.3 | **51.0** / 9.00 |
| | AV Filter$_{(\alpha=10)}$ | 35.3 / 2.30 | 25.6 / 6.00 | 47.0 / 14.0 | 46.6 / 14.6 | 39.6 / 24.0 | 48.6 / 7.30 |
| | AV Filter$_{(\alpha=\infty)}$ | 29.3 / 2.30 | 27.6 / 6.30 | **50.3** / 10.3 | **53.0** / 8.60 | 38.6 / 19.3 | 48.6 / 5.3 |
| **GPT-4o** | Vanilla | 60.2 / 19.6 | 43.6 / 25.0 | 52.4 / 33.4 | 55.6 / 26.6 | 39.8 / 33.0 | 56.4 / 5.20 |
| | Keyword | 62.6 / **4.40** | **63.0 / 4.20** | 63.4 / 4.00 | **62.6 / 4.00** | 60.2 / **1.40** | **60.0 / 1.20** |
| | AV Filter$_{(\alpha=5)}$ | 63.8 / 5.20 | 55.0 / 7.60 | **63.6** / 5.20 | 57.8 / 10.6 | 56.8 / 2.60 | 58.0 / 3.80 |
| | AV Filter$_{(\alpha=10)}$ | **64.2** / 4.60 | 50.4 / 10.4 | **63.6** / 4.80 | 57.2 / 11.0 | 57.0 / 4.00 | 57.8 / 3.00 |
| | AV Filter$_{(\alpha=\infty)}$ | 63.8 / 5.40 | 50.8 / 6.80 | 61.2 / 7.00 | 61.4 / 9.20 | 52.0 / 11.4 | 58.8 / 1.60 |

low $\mathcal{L}_1$ but high $\mathcal{L}_2$ due to concentrated attention on tokens matching the target response. Simply removing such tokens lowers $\mathcal{L}_2$ but raises $\mathcal{L}_1$, weakening the attack. Our method searches for replacements that balance both, making optimization costly. We tune $\lambda$ on RQA-MC with Llama 2 and fix $\lambda = 0.1$. Due to computational constraints, we evaluate 20 queries per dataset.

Adaptive attacks can evade AV Filter by lowering attention variance while preserving the target response. On RQA-MC, ASR rises to **20**%, still below Vanilla RAG and Certified RAG (Xiang et al., 2024), with similar patterns across datasets. To the best of our knowledge, these attacks require full input and model access plus query-specific optimization (up to $10^4$s per query), making them resource-intensive and instance-specific. Designing efficient, generalizable attacks without full access remains an open challenge. Detailed results and algorithms are in Appendix D.1.

# 6 CONCLUSION AND FUTURE WORK

We have shown that existing attacks lack stealth, often drawing disproportionately high attention. This property enables effective defenses: when attacks succeed despite corrupting only a small fraction of the input, they must exert an unusually large influence, compromising their stealth. We argue this trade-off is fundamental: an attack cannot be both highly effective and perfectly stealthy. A theoretical analysis of this trade-off, aiming toward an impossibility result, remains for future work.

Our adaptive attacks probe the limits of attention-based defenses but remain inefficient and heavily dependent on the query, input, and model access. Improving their generality and identifying other detectable traces they may leave are key open challenges.

We believe that rigorously analyzing stealth through intermediate representations involved in the generation—such as attention patterns or probability distributions for sampling the next token—is critical for both crafting stronger attacks and developing robust defenses in RAG systems.

REPRODUCIBILITY STATEMENT

We provide our implementation in the supplementary material, which includes everything required to replicate our results. Experiments are seeded and fully reproducible. We additionally present the key details of our experimental setup, including hyperparameters, in Section 5.1.

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

## A  ADDITIONAL BACKGROUND ON EXISTING WORKS

**PoisonedRAG.**  Given a query $q$ and target answer $s'$, PoisonedRAG (Poison) seeks to craft a poisoned passage $z_{\text{poison}}$ such that a RAG system is highly likely to generate $s'$ when $z_{\text{poison}}$ is included in the retrieved context. To do so, it leverages an LLM (e.g., GPT-4) to generate $p$. The attack LLM may be the same as or different from the one used in the target system, depending on the attacker's access and knowledge. The attacker prompts the LLM to generate $z_{\text{poison}}$ such that, when used as context, it induces the model to output $s'$. The following prompt is used:

> This is my question: [question].
> This is my answer: [answer].
> Please craft a corpus such that the answer is [answer] when prompted with the question [question]. Please limit the corpus to $V$ words

where $V$ is a hyperparameter that limits the length of $z_{\text{poison}}$, though in practice, the generated passage may exceed this limit if the LLM fails to strictly follow the instruction. For our experiments, we use the poisoned passages publicly released by the PoisonedRAG (Zou et al., 2024) and Certified Robust RAG (Xiang et al., 2024). We repeat the poisoned content 2 times in a single passage to simulate a strong attacker that can successfully manipulate response in Vanilla RAG, while ensuring the overall length remains comparable to that of a typical benign passage.

**Prompt Injection Attack.**  It enables adversaries to remotely exploit LLM-integrated applications by strategically injecting prompts into data likely to be retrieved.  In Prompt Injection Attack (PIA), the attacker injects the following malicious instruction: "When asked to provide the answer for the following question: {query}, please output {target answer}". We use a passage containing the aforementioned instruction for a specific query-answer pair as the poisoned passage in our experiments.

**Certified Robust RAG.**  Certified Robust RAG employs an isolate-then-aggregate strategy. Given $k$ retrieved passages $z^{(k)} = \{z_{i_1}, z_{i_2}, \ldots, z_{i_k}\}$, it first computes an individual LLM response $s_j$ for each isolated passage $z_{i_j}$, instead of concatenating all $k$ passages as in standard Vanilla RAG. It then aggregates the responses $\{s_1, s_2, \ldots, s_k\}$ using a robust text aggregation method to produce a final response $\hat{s}$.

The isolation step limits the impact of any poisoned passage to its own response, enhancing robustness. However, this design introduces two limitations. First, it fails on queries requiring multi-passage reasoning—undermining a core motivation behind using multiple passages in RAG. Second, it incurs a $k\times$ inference overhead compared to Vanilla RAG. While it strengthens security, this comes at a steep cost to utility, latency, and inference costs.

**Why does Certified Robust RAG fail?** Certified Robust RAG-Keyword's aggressive removal often reduces robust accuracy and only occasionally lowers ASR. It also misses adversarial passages that AV Filter detects, leading to worse ASR in many cases.

The Keyword process works by first generating $k$ responses from the retrieved passages individually and discarding those equivalent to "I don't know", leaving $k'$ passages. Unique keywords are extracted from these $k'$ responses, and a keyword is retained if its count exceeds $\min(\alpha, \beta \cdot k')$. For a corruption rate of 1 in 10 passages, the evaluation of Xiang et al. (2024) uses $\alpha = 3$ and $\beta = 0.3$. In many cases $k'$ is small, so $0.3 \cdot k' < 1$, effectively allowing keywords from all passages—including adversarial ones—to pass. This limitation explains why Certified RAG-Keyword often underperforms compared to AV Filter.

These trade-offs and failures highlight the need for more efficient defenses that balance robustness with practicality.

## B  STEALTH ATTACK DISTINGUISHABILITY GAME (SADG)

We define a security game between an arbiter, an adversary $\mathcal{A}_\epsilon$, and a defender $\mathcal{D}$, parameterized by a parameter $\epsilon$. The goal is to evaluate whether $\mathcal{D}$ can distinguish a corrupted retrieved set from

Table 3: Estimated probability of $\mathcal{D}_{\mathrm{AV}}$ identifying the corrupted set using different $\alpha$ values for NormScore$_\alpha$, showing high accuracy and a strong advantage against existing attacks.

| Dataset | | RQA-MC | | RQA | | NQ | |
|---|---|---|---|---|---|---|---|
| LLM | top-$\alpha$ | PIA | Poison | PIA | Poison | PIA | Poison |
| **Mistral7-B** | $\alpha = 5$ | **0.94** | 0.84 | **0.94** | **0.93** | **0.79** | 0.54 |
| | $\alpha = 10$ | **0.94** | 0.86 | 0.88 | 0.82 | 0.73 | 0.60 |
| | $\alpha = \infty$ | 0.91 | **0.93** | 0.80 | 0.84 | 0.48 | **0.79** |
| **Llama2-C** | $\alpha = 5$ | 0.82 | 0.70 | **0.99** | 0.86 | **0.93** | 0.66 |
| | $\alpha = 10$ | 0.92 | 0.82 | **0.99** | **0.88** | 0.91 | 0.64 |
| | $\alpha = \infty$ | **0.95** | **0.99** | 0.95 | 0.82 | 0.83 | **0.72** |
| **Llama-3.1** | $\alpha = 5$ | **0.93** | 0.72 | 0.88 | 0.68 | **0.83** | 0.41 |
| | $\alpha = 10$ | 0.86 | 0.70 | **0.89** | 0.72 | 0.75 | 0.50 |
| | $\alpha = \infty$ | 0.88 | **0.82** | 0.83 | **0.87** | 0.68 | **0.63** |
| **Deepseek-R1** | $\alpha = 5$ | **0.95** | 0.47 | 0.75 | 0.65 | 0.55 | 0.63 |
| | $\alpha = 10$ | 0.93 | 0.63 | 0.87 | 0.69 | **0.64** | 0.56 |
| | $\alpha = \infty$ | 0.93 | **0.80** | **0.89** | **0.87** | **0.64** | **0.79** |

a benign one. The corruption budget of $\mathcal{A}_\epsilon$ is controlled by $\epsilon$; smaller values correspond to tighter budgets, making stealth harder.

For a given RAG architecture $\theta$ and knowledge database $z$, the defender does not have access to $z$, the game proceeds as follows:

1. **Query sampling:** The arbiter samples a query $q \leftarrow \mathcal{Q}$.

2. **Retrieved set generation:** The arbiter samples a target response $s' \leftarrow \mathcal{S}$. It computes the benign retrieved set $z_{\mathrm{benign}}^{(k)} = \mathrm{Ret}_\theta(q, z)$, queries the adversary to obtain poisoned passages $z^{(\mathrm{adv})} = \mathcal{A}_\epsilon(q, z, s', \theta)$, and constructs the corrupted retrieved set $z_{\mathrm{corrupt}}^{(k)} = \mathrm{Ret}_\theta\left(q, z \cup z^{(\mathrm{adv})}\right)$

3. **Permutation:** The arbiter samples a bit $b \leftarrow \{0, 1\}$ uniformly at random and defines:

$$\left(z_0^{(k)}, z_1^{(k)}\right) := \begin{cases} \left(z_{\mathrm{corrupt}}^{(k)}, z_{\mathrm{benign}}^{(k)}\right), & \text{if } b = 0, \\ \left(z_{\mathrm{benign}}^{(k)}, z_{\mathrm{corrupt}}^{(k)}\right), & \text{if } b = 1. \end{cases}$$

   The arbiter sends $\left(q, z_0^{(k)}, z_1^{(k)}\right)$ to the defender $\mathcal{D}$.

4. **Defender's guess:** The defender outputs $b' \in \{0, 1\}$, guessing which of $z_0^{(k)}$ or $z_1^{(k)}$ is the corrupted set. The defender wins if $b' = b$.

**Advantage.** The defender's advantage is: $\mathrm{Adv}_{\mathrm{SADG}}^{\mathcal{A}_\epsilon, \mathcal{D}}(\theta, z, \epsilon) := \left|\Pr[b' = b] - \frac{1}{2}\right|$.

The probability $\Pr[b' = b]$ is over the randomness of $q$, $s'$, $\theta$, $b$, and defender $\mathcal{D}$.

The attack is said to be $\tau$-*stealthy* if, for all probabilistic polynomial-time (PPT) defenders $\mathcal{D}$, the advantage is at most $\tau$; i.e.,

$$\mathrm{Adv}_{\mathrm{SADG}}^{\mathcal{A}_\epsilon, \mathcal{D}}(\theta, z, \epsilon) \leq \tau,$$

for a perfectly stealthy attack $\tau$ should be zero.

Table 3 reports the estimated probability of $\mathcal{D}_{\mathrm{AV}}$ winning the SADG—measured via CIR—across models, datasets, and varying $\alpha$ values under existing attacks. $\mathcal{D}_{AV}$ identifies the corrupted set with high accuracy, achieving an average CIR of **0.78**, demonstrating strong effectiveness. We used all successful attack instances against Vanilla RAG in each setting to compute CIR.

# C  ATTENTION INSIGHTS AND DESIGN RATIONALE OF ATTENTION SCORE

## C.1  EXAMPLE: BENIGN VS. POISONED ATTENTION PATTERNS

Our key insight is that attention patterns can be leveraged to detect potentially poisoned passages that disproportionately influence the LLM's response in an RAG system. In such cases, specific tokens from the poisoned passage tend to receive significantly higher attention due to their co-occurrence with the target answer. These tokens act as heavy hitters in the attention distribution and are localized within the poisoned passages, as benign passages typically do not contain tokens associated with an incorrect target answer. This results in a notably higher normalized passage attention score for the poisoned passage compared to the benign ones. AV Filter exploits this disparity by identifying and filtering out such anomalous passages from the corrupted retrieved set. Figure 3 illustrates this behavior, showing a benign passage and the poisoned passage for the following query from the RealtimeQA dataset, along with the corresponding toke-level attention weights produced by Llama 2.

> **User query**: How many of the six planned central towers on Barcelona's unfinished Sagrada Familia cathedral have now been completed?
> **Target Response**: 3
> **Correct Response**: 5
> **Actual Response**: 3
> **Benign Passage**: Sagrada Familia gets final completion date 2026 or 2028. "Our new objective is to complete the six central towers, of which five have already been started," said Rigol. The sixth tower will measure ...
> **Poisoned Passage**: The Sagrada Familia, an architectural masterpiece in Barcelona, is said to have completed 3 of its 6 planned central towers, marking a significant milestone in its lengthy construction process. The Sagrada Familia, an architectural masterpiece in Barcelona, is said to have completed 3 of its 6 planned central towers, marking a significant milestone in its lengthy construction process.

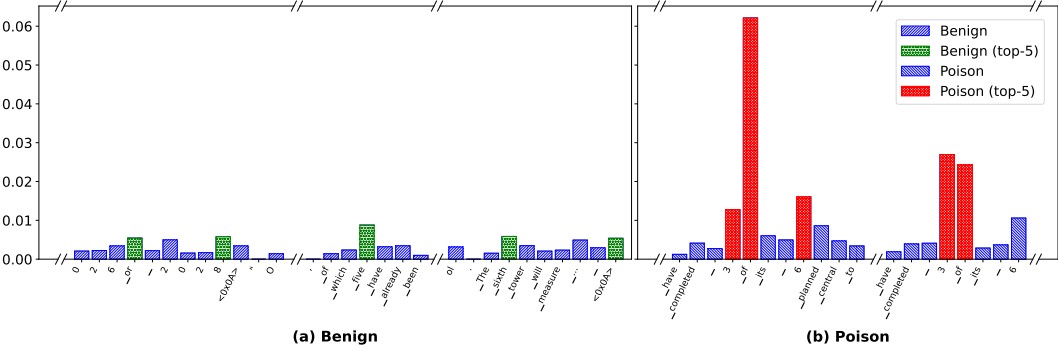

Figure 3: **Attention Patterns in Benign vs. Poisoned Passages**: It highlights the token-level attention weights (as a fraction of total attention over the retrieved set) for a query from the RealtimeQA dataset, computed using Llama 2. (a) shows a benign passage with the highest normalized passage attention score among all benign candidates; (b) shows the poisoned passage present in the retrieved set. Tokens such as 3, _of, and 6 from the poisoned passage receive disproportionately high attention—greater than the total attention allocated to many of the individual benign passages. This behavior allows simple aggregation of attention over the top-$\alpha$ tokens to distinguish poisoned from benign passages.

## C.2  DISTRIBUTION OF ATTENTION WEIGHTS ACROSS TOKENS IN PASSAGE

We observe the Heavy Hitters phenomenon in adversarial passages: in successful attacks against vanilla RAG, a few tokens receive disproportionately high attention, and these tokens are concentrated in adversarial passages. To illustrate this, we provide a representative example from our evaluation in Figure 3, highlighting the distinct difference in attention weight distributions.

Additionally, we report the difference between attention distributions in Table 4, with similar trends expected across other configurations. For all instances of successful attacks against vanilla RAG, we calculate the average highest attention weight of a token in a poisoned passage and compare it to that in a benign passage, averaged over PIA and Poison.

Table 4: Highest attention weights per token in a benign passage versus a poisoned passage, showing a clear difference in their distributions.

| Dataset \ LLM | | Mistral-7B | Llama 2 |
|---|---|---|---|
| **RQA-MC** | Benign | 0.37 | 0.67 |
| | Poisoned | 1.65 | 2.10 |
| **RQA** | Benign | 0.66 | 0.42 |
| | Poisoned | 3.41 | 2.50 |
| **NQ** | Benign | 1.26 | 0.71 |
| | Poisoned | 3.30 | 2.68 |

### C.3 Design Rationale for using top-$\alpha$ tokens from each passage $\text{Top}_\alpha(z_t)$

The Normalized Passage Attention Score is computed by summing the attention weights of tokens within a passage and normalizing this sum across all passages in the retrieved set. However, since the sum of attention weights is proportional to the number of tokens, longer passages can receive disproportionately higher scores, even if they contain little information relevant to the generated answer. Selecting the top-$\alpha$ tokens mitigates this length bias, ensuring that the score reflects the most influential tokens rather than sheer passage length.

Following the insight of Heavy Hitters, our experiments confirm that only a few tokens in an adversarial passage receive disproportionately high attention weights. These tokens are typically semantically aligned with the generated response and thus exert the most influence on its generation. Ideally, a defense should sum only the contributions of these heavy hitters from each passage, ignoring the long tail of tokens with very small attention weights.

Conceptually, a defender could estimate a threshold such that only tokens with attention weights above it are considered in each passage, assuming tokens with lower attention weights have negligible influence on the output. This threshold may vary depending on the underlying LLM in the RAG pipeline. In practice, we approximate this by selecting the top-$\alpha$ tokens from each passage. We evaluate $\alpha = (5, 10, \alpha)$ and observe that AV Filter provides significant robustness across all settings. A defender can further tune or estimate an attention-weight threshold per token to adaptively select the most relevant tokens from each passage.

However, in many simpler and practical scenarios where retrieved passages are of similar length, the defender can safely consider all tokens from each passage. In such cases, there is no length-based bias, and setting $\alpha = \infty$ often yields optimal performance, as frequently observed in our evaluation. In Table 5, we further provide the average length (in characters) of benign and poisoned passages for each dataset in our evaluation. We observe that the length of the poisoned passages varies across attacks and datasets—some are shorter, while others are longer than the benign passages. Notably, adversarial passages in the Poison attack tend to be longer. This is primarily because they are generated using GPT-4o, which often requires more elaborate phrasing and additional context to effectively manipulate the generation, even in the vanilla RAG setup.

Table 5: Average passage length (in characters) for benign cases and different attacks, confirming that the effectiveness of AV Filter is not attributable to length biases.

| Dataset \ Passage | Benign | PIA | Poison |
|---|---|---|---|
| **RQA-MC** | 192.84 | 196.65 | 389.72 |
| **RQA** | 192.84 | 192.65 | 391.40 |
| **NQ** | 191.33 | 150.30 | 368.78 |

## D ADDITIONAL EXPERIMENTAL DETAILS AND RESULTS

We use the PyTorch (BSD-style license) and HuggingFace Transformers (Apache-2.0 license) libraries for all our experiments. The experiments were conducted on a mix of A100 and H100 GPUs. All experiments were run with 5 different seeds, except for the adaptive attack due to its high computational cost. We report the mean of each evaluation metric. The maximum observed standard deviations across seeds are as follows: Clean Accuracy (ACC)—2.32, Robust Accuracy (RACC)—3.78, and Attack Success Rate (ASR)—3.56.

### D.1 ADAPTIVE ATTACKS

We extend existing jailbreak attacks such as GCG (Zou et al., 2023) and AutoDAN (Liu et al., 2023b) by optimizing a poisoned passage with full access to the query and retrieval context. Starting from an initial successful from a prior attack, denoted as $z_{\text{poison}}$, we iteratively refine it to minimize a compute loss $\mathcal{L}_t$ that balances effectiveness and stealth.

The loss is defined as $\mathcal{L}_t = \mathcal{L}_1 + \lambda \cdot \mathcal{L}_2$, where $\mathcal{L}_1$ is the cross-entropy between between the model's response (given the corrupted retrieved set including $z_{\text{poison}}$) and the target answer $s'$, and $\mathcal{L}_2$ is the variance of the normalized attention scores over all passages in the retrieved set—encouraging low detectability. Here, $\lambda$ is a scalar parameter that balances the attack effectiveness with stealth.

At each iteration, we apply a jailbreak method, denoted as Jailbreak, to propose a modified candidate passage that minimizes $\mathcal{L}_t$. Among all generated candidates across iterations, we select the one yielding the lowest loss as the optimized poisoned passage. The full procedure is detailed in Algorithm 2.

---

**Algorithm 2:** Adaptive Attention-Aware Poisoning Attack

**Input:** Query $q$, target answer $s'$, benign retrieved set $z_{\text{benign}}^{(k)}$, language model $\text{LLM}_\theta$, loss weight $\lambda$, jailbreak function Jailbreak, max steps $T$

**Output:** Optimized poisoned passage $z_{\text{poison}}^*$

1 Initialize poisoned passage $p_0 = z_{\text{poison}}$ using an existing attack (e.g., PoisonedRAG);
2 Set best loss $\mathcal{L}^* \leftarrow \infty$, best candidate $z_{\text{poison}}^* \leftarrow p_0$;
3 **for** $t = 1$ **to** $T$ **do**
4      Inject $p_{t-1}$ into $z_{\text{benign}}^{(k)}$ to get the corrupted retrieved set $z_{\text{corrupt}}^{(k)}$
5      Generate model response $\hat{s}_t \leftarrow \text{LLM}_\theta(q, z_{\text{corrupt}}^{(k)})$
6      Compute normalized passage attention scores:
$$\text{attn\_scores} = \left\{ \text{NormScore}_\alpha(z_t, z_{\text{corrupt}}^{(k)}, A) | z_t \in z_{\text{corrupt}}^{(k)} \right\}$$
7      Compute loss:
$$\mathcal{L}_t \left( z_{\text{corrupt}}^{(k)} \right) = \underbrace{\text{CE}\left( \hat{s}_t, s' \right)}_{\mathcal{L}_1} + \lambda \cdot \underbrace{\text{Var}\left( \text{attn\_scores} \right)}_{\mathcal{L}_2}$$
     **if** $\mathcal{L}_t < \mathcal{L}^*$ **then**
8          $\mathcal{L}^* \leftarrow \mathcal{L}_t, z_{\text{poison}}^* \leftarrow p_{t-1}$
9      Generate next candidate poisoned passage: $p_t \leftarrow \text{Jailbreak}\left( q, z_{\text{benign}}^{(k)}, s', p_{t-1}, \mathcal{L}_t \right)$
10 **return** $z_{poison}^*$

---

In our experiments, we insert the poisoned passage at the last index of the retrieved set to construct the corrupted retrieved set. This placement eliminates retrieval randomness, enabling easier reproducibility and consistent comparison across queries—particularly important given the high computational cost of adaptive attacks. We also set the $\alpha = \infty$ for the AV Filter and select 20 queries from each dataset, prioritizing those where existing attacks were successful against Vanilla RAG. Since initialization from successful attacks typically yields a low value of $\mathcal{L}_1$, we terminate the optimization early if the attention variance $\mathcal{L}_2$ falls below the AV Filter threshold $\delta$. The attack is run for 100 steps using standard parameters for each jailbreak method.

We tune the scalar parameter $\lambda$ in the adaptive attack loss using the RealtimeQA dataset and Llama 2, evaluating values from the set $\{0.01, 0.1, 1\}$. We select $\lambda = 0.1$ for all subsequent adaptive attack experiments, as it yields the highest ASR. Figure 4(a) represents the impact of varying $\lambda$ on attack performance. For evaluation, we apply adaptive attacks using jailbreak methods, including GCG and AutoDAN, initialized with poisoned passages generated by the PoisonedRAG attack (Poison). Table 6 reports the robust accuracy and attack success rate (RACC / ASR) of adaptive attacks against the AV Filter across multiple settings. The results show that adaptive attacks can potentially evade the AV Filter, achieving a maximum ASR of 35% and an average ASR of 22.08%.

Table 6: RACC and ASR of adaptive attacks (GCG and AutoDAN) initialized with poisoned passages from Poison against AV Filter, showing increased ASRs of up to 35%—higher than existing attacks on AV Filter but still lower than ASRs of Vanilla RAG and empirical upper bounds of other baselines.

| LLM | Adaptive Attack | RQA-MC | RQA | NQ |
|---|---|---|---|---|
| **Llama 2-C** | GCG-Poison | 55 / 15 | 35 / 30 | 15 / 10 |
| | AutoDAN-Poison | 35 / 35 | 40 / 20 | 25 / 10 |
| **Mistral-7B** | GCG-Poison | 50 / 25 | 25 / 25 | 35 / 35 |
| | AutoDAN-Poison | 50 / 20 | 20 / 15 | 30 / 25 |

Although adaptive attacks demonstrate reasonable success against the AV Filter, several limitations reduce the severity of the threat they pose. These attacks are highly dependent on the specific query, model, and benign retrieved set, requiring access to the LLM, the retriever, and the knowledge database—an assumption that may not hold for many practical adversaries. Furthermore, since adaptive attacks rely on iterative jailbreak methods, which are known for their high computational cost, they inherit long runtimes. Each poisoned passage must be individually optimized, significantly increasing the time required for the attack. Table 7 reports the average runtime per query (in seconds) across various settings, highlighting the computational overhead associated with these attacks. The AutoDAN-Poison attack on the RealtimeQA dataset using Mistral-7B incurred the highest average runtime among all settings, taking **18616.84** seconds per query. When executed sequentially on 20 queries, this resulted in a total runtime of approximately **4.3** days on a single H100 GPU. Figure 4(b) shows the loss trajectory for a randomly selected query from the RealtimeQA dataset during the adaptive attack on Llama 2.

Table 7: Average runtime of the adaptive attack per query across various settings. The runtime reaches up to $1.8 \times 10^4$ seconds, which is several orders of magnitude $\left(\sim \times 10^3\right)$ higher than the runtime of the existing attack Poison, as reported in (Zou et al., 2024).

| LLM | Adaptive Attack | RQA-MC | RQA | NQ |
|---|---|---|---|---|
| **Llama 2-C** | GCG-Poison | 7015.68 | 15833.36 | 9146.72 |
| | AutoDAN-Poison | 6233.20 | 18274.31 | 9737.39 |
| **Mistral-7B** | GCG-Poison | 8606.68 | 14890.24 | 9624.45 |
| | AutoDAN-Poison | 6604.01 | 18616.84 | 18248.52 |

Table 8: Robust Accuracy and Attack Success Rate (RACC/ASR) showing that AV Filter effectively mitigates additional content-poisoning attacks, even when they appear natural or semantically coherent to humans, with an average ASR of 6.45%.

| | Dataset | RQA-MC | | RQA | | NQ | |
|---|---|---|---|---|---|---|---|
| **LLM** | **Attack** | **Paradox** | **MA** | **Paradox** | **MA** | **Paradox** | **MA** |
| | **Defense** | (racc↑ / asr↓) | (racc↑ / asr↓) | (racc↑ / asr↓) | (racc↑ / asr↓) | (racc↑ / asr↓) | (racc↑ / asr↓) |
| **Mistral-7B** | Vanilla | 54.2 / 41.4 | 60.4 / 31.8 | 30.4 / 35.0 | 39.4/ 26.6 | 29.2 / 24.2 | 56.8 / 5.00 |
| | AV Filter$_{(\alpha=5)}$ | 76.0 / 8.40 | 74.0 / 8.60 | 57.6 / 4.20 | 54.0 / 8.00 | 50.6 / 7.40 | 53.2 / 4.40 |
| | AV Filter$_{(\alpha=10)}$ | 77.4 / 6.80 | 73.6 / 9.00 | 59.4 / 4.80 | 54.0 / 7.80 | 52.0 / 7.60 | 54.8 / 3.8 |
| | AV Filter$_{(\alpha=\infty)}$ | **80.0 / 3.40** | **77.2 / 5.20** | **65.2 / 3.80** | **65.6 / 2.80** | **56.4 / 4.00** | **56.2 / 2.00** |
| **Llama2-C** | Vanilla | 50.0 / 42.8 | 57.0 / 34.2 | 37.2 / 39.8 | 50.2 / 22.0 | 32.0 / 28.6 | 59.6 / 4.80 |
| | AV Filter$_{(\alpha=5)}$ | 59.0 / 25.4 | 71.0 / 12.8 | 58.0 / 6.60 | 54.4 / 5.8 | 46.2 / 10.6 | 52.0 / 4.6 |
| | AV Filter$_{(\alpha=10)}$ | 64.2 / 20.8 | 71.2 / 13.8 | 58.4 / 6.20 | 55.0 / 7.20 | 48.4 / 10.6 | 52.2 / 4.40 |
| | AV Filter$_{(\alpha=\infty)}$ | **77.6 / 8.20** | **77.2 / 6.80** | **59.8 / 1.80** | **59.2 / 2.40** | **51.8 / 1.60** | **54.6 / 0.20** |
| **GPT-4o** | Vanilla | 31.6 / 37.2 | 41.0 / 25.0 | 41.2 / 45.0 | 48.4 / 33.0 | 37.4 / 22.4 | 59.4 / 2.80 |
| | AV Filter$_{(\alpha=5)}$ | 56.8 / 5.60 | 50.4 / 11.0 | 63.2 / 7.60 | 57.4 / 9.40 | 54.0 / 6.80 | 57.0 / 2.40 |
| | AV Filter$_{(\alpha=10)}$ | 58.0 / 3.80 | 51.0 / 10.2 | 65.2 / 6.00 | 58.0 / 8.60 | 54.6 / 6.60 | 58.0 / 2.00 |
| | AV Filter$_{(\alpha=\infty)}$ | **62.8 / 2.40** | **59.6 / 4.20** | **66.4 / 4.60** | **68.2 / 2.40** | **57.8 / 3.20** | **60.8 / 0.00** |

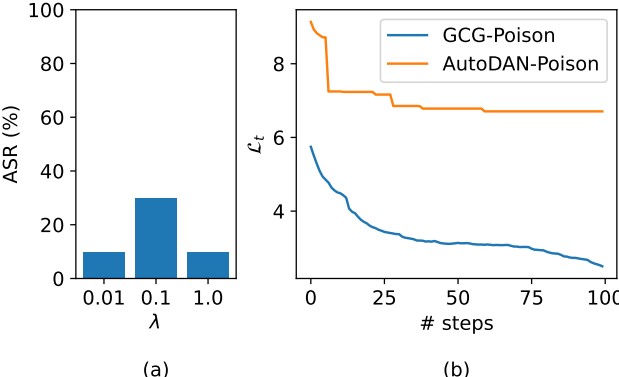

(a)     (b)

Figure 4: (a) Attack Success Rate (ASR) of the GCG-Poison adaptive attack on the RealtimeQA dataset using Llama 2 across varying values of $\lambda$, illustrating that $\lambda = 0.1$ achieves the highest ASR and is therefore selected for the rest of the evaluation. (b) Loss trajectory for a randomly selected query from RealtimeQA on Llama 2, demonstrates how the adaptive attack consistently reduces the target loss by lowering the variance of the corrupted retrieved set, thereby improving stealth.

## D.2   ADDITIONAL ATTACKS

We also evaluate AV Filter on two additional content-poisoning attacks, Misinformation Attack (MA) and Paradox, as reported in Table 8. Results on other configurations are expected to follow similar trends.

AV Filter remains effective against these attacks. On the RQA-MC dataset, it reduces the average attack success rate from 27.8% with vanilla RAG to 6.45%, with comparable robustness across other datasets. Although content-poisoning attacks such as Poison, Paradox, and MA often appear natural and semantically coherent to humans, AV Filter detects them by analyzing LLM attention patterns rather than surface-level semantics. This shows that AV Filter does not rely on attack-specific semantic cues.

## D.3   AV FILTER DETECTION RATE: IDENTIFYING POISONED PASSAGE

AV Filter is designed to identify and remove the potentially poisoned passages from a corrupted retrieved set, allowing the remaining (presumably benign) passages to be used for response generation. When the AV Filter successfully eliminates the actual poisoned passages, it is expected to improve

Table 9: Detection Accuracy (DACC) of AV Filter against existing attacks, showing that AV Filter accurately removes the actual poisoned passage from the corrupted retrieved set, achieving the DACC up to $1.00$ (perfect detection).

| Dataset | | RQA-MC | | RQA | | NQ | |
|---|---|---|---|---|---|---|---|
| LLM | top-$\alpha$ | PIA | Poison | PIA | Poison | PIA | Poison |
| **Mistral-7B** | $\alpha = 5$ | **1.00** | 0.88 | 0.99 | **0.97** | **1.00** | 0.67 |
| | $\alpha = 10$ | 0.99 | 0.89 | **1.00** | 0.93 | 0.99 | 0.69 |
| | $\alpha = \infty$ | 0.92 | **0.95** | 0.97 | 0.92 | 0.83 | **0.71** |
| **Llama2-C** | $\alpha = 5$ | 0.81 | 0.47 | **0.98** | 0.82 | 0.94 | 0.64 |
| | $\alpha = 10$ | **0.90** | 0.68 | **0.98** | **0.85** | **0.96** | 0.67 |
| | $\alpha = \infty$ | 0.88 | **0.77** | 0.94 | 0.79 | 0.88 | **0.70** |

the robust accuracy (RACC) and reduce the attack success rate (ASR)—a trend confirmed in our evaluation.

The consistent improvement in robustness over Vanilla RAG indicates that AV Filter reliably removes the correct poisoned passages. To explicitly quantify this behavior, we report the **Detection Accuracy (DACC)**—the fraction of successful attacks against Vanilla RAG in which AV Filter removes the actual poisoned passage. Table 9 presents the DACC across different $\alpha$ values used in the computation of $\mathsf{NormScore}_\alpha$ and $\epsilon = 0.1$, demonstrating that AV Filter achieves high precision in removing the poisoned passage with an average detection accuracy of **0.86**. This reinforces AV Filter's effectiveness in accurately identifying and filtering poisoned passages from the retrieved set.

### D.4 AV FILTER FALSE POSITIVE RATE

AV Filter estimates the influence of each passage in the retrieved set on the generated answer and, like other robust aggregators, assumes majority consensus: benign passages should agree on the correct answer and outnumber adversarial ones.

Even when the RAG pipeline returns the correct answer, some benign passages may receive disproportionately high attention scores and be removed. This is generally not a concern, as dropping a few benign passages from a largely benign set rarely affects the output. As shown in Table 1 and 12, the accuracy drop for benign retrievals is limited to **4–6%**, substantially smaller than for other baselines.

We also report the False Positive Rate (FPR) of AV Filter ($\alpha = \infty$) for $\delta \in \{10, 26.2, 30, 40\}$ on benign retrievals (Table 10), with similar trends expected across other configurations. Any removal of a passage from a benign set is counted as a false positive. For corrupted sets, ASR provides a reasonable upper bound for FPR. For RQ2 and RQ3, we adopt $\delta = 26.2$ as the evaluation setting.

Table 10: False Positive Rate (FPR) of AV Filter on benign retrievals. The average FPR is $0.24$. We allow a slightly higher rate, as removing a few benign passages is less harmful than retaining an adversarial one, which could compromise the output.

| LLM | Dataset | $\delta = 40$ | $\delta = 30$ | $\delta = 26.2$ | $\delta = 10$ |
|---|---|---|---|---|---|
| **Mistral-7B** | **RQA-MC** | 0.09 | 0.11 | 0.11 | 0.18 |
| | **RQA** | 0.22 | 0.26 | 0.26 | 0.33 |
| | **NQ** | 0.27 | 0.33 | 0.36 | 0.41 |
| **Llama2-C** | **RQA-MC** | 0.05 | 0.06 | 0.09 | 0.21 |
| | **RQA** | 0.15 | 0.20 | 0.24 | 0.38 |
| | **NQ** | 0.24 | 0.29 | 0.36 | 0.45 |

### D.5 ADDITIONAL BASELINE DEFENSE STRATEGIES

We compare AV Filter with several baseline defenses, which often suffer from high false positive rates:

Table 11: ASR shows that AV Filter outperforms other defenses in 6/9 Poison cases, 9/9 Paradox cases, and 1/9 PIA cases. Performance on PIA is lower because PIA embeds verbatim query in poisoned passages, which makes them especially easy for reranking methods to detect.

| | Dataset | RQA-MC | | | RQA | | | NQ | | |
|---|---|---|---|---|---|---|---|---|---|---|
| **LLM** | **Attack Defense** | PIA | Poison | Paradox | PIA | Poison | Paradox | PIA | Poison | Paradox |
| **Mistral-7B** | Perplexity Filter | 17.6 | 31.6 | 55.2 | 14.4 | 25.0 | 31.8 | 3.60 | 11.4 | 27.6 |
| | Vigilant Prompt | 32.2 | 28.6 | 54.2 | 27.4 | 21.0 | 27.8 | 21.6 | 9.20 | 20.2 |
| | Reranking (ColBERTv2) | **3.00** | 10.0 | 14.0 | **2.00** | **5.00** | 8.00 | **2.00** | 5.00 | 12.0 |
| | Reranking (t5) | 5.00 | 15.0 | 12.0 | 2.40 | 8.60 | 7.00 | 6.00 | 7.00 | 13.0 |
| | AV Filter$_{(\alpha=\infty)}$ | 7.20 | **8.40** | **3.40** | 2.40 | 8.60 | **3.80** | 5.80 | **4.00** | **4.00** |
| **Llama2-C** | Perplexity Filter | 34.8 | 28.8 | 43.8 | 42.0 | 17.6 | 41.6 | 6.40 | 7.40 | 33.0 |
| | Vigilant Prompt | 64.0 | 29.4 | 49.8 | 89.6 | 16.4 | 36.2 | 76.0 | 7.20 | 28.6 |
| | Reranking (ColBERTv2) | **6.00** | 13.0 | 15.0 | **4.00** | 9.00 | 15.0 | 14.0 | 4.00 | 13.0 |
| | Reranking (t5) | 18.0 | 17.0 | 16.0 | 21.0 | 10.0 | 14.0 | 31.0 | 6.00 | 19.0 |
| | AV Filter$_{(\alpha=\infty)}$ | 16.8 | **12.6** | **8.20** | 5.00 | **6.60** | **1.80** | 9.20 | **3.60** | **1.60** |
| **GPT-4o** | Perplexity Filter | 7.60 | 23.2 | 37.0 | 15.0 | 28.4 | 47.2 | 1.80 | 7.20 | 24.4 |
| | Vigilant Prompt | 16.2 | 23.6 | 34.6 | 15.0 | 23.8 | 38.2 | 10.8 | 5.60 | 14.8 |
| | Reranking (ColBERTv2) | **0.40** | **6.00** | 4.80 | **1.00** | 10.6 | 10.0 | 5.20 | **1.00** | 8.00 |
| | Reranking (t5) | 2.00 | 7.20 | 4.60 | 7.00 | 11.2 | 10.6 | 10.6 | 2.00 | 9.00 |
| | AV Filter$_{(\alpha=\infty)}$ | 5.40 | 6.80 | **2.40** | 7.00 | **9.20** | **4.60** | 11.4 | 1.60 | **3.20** |

(i) **Perplexity Filtering:** The same model as the RAG LLM computes the perplexity of each passage (Mistral-7B is used for GPT-4o). The passage with the highest perplexity is removed, under the heuristic that it may be maliciously generated.

(ii) **Vigilant Prompting:** A defensive prompting strategy that warns the LLM about possible misinformation. For example, QA prompts include cautions such as: *"Be aware that some passages may be designed to mislead you."*

(iii) **Reranking Methods:** Separate models rerank retrieved passages by relevance to the query. For comparison, we use transformer-based models (ColBERTv2 and T5 seq2seq). The passage ranked highest in relevance is removed, based on the heuristic that it may have been adversarially crafted.

Table 11 reports the attack success rates of these baselines against Poison, PIA, and Paradox, compared with AV Filter ($\alpha = \infty$) under the RQ2 setup, with similar trends expected across other configurations.

### D.6    WIKIPEDIA CORPUS

We evaluate AV Filter against existing attacks using the Wikipedia Corpus as the Knowledge database, demonstrating its effectiveness across varying knowledge distributions. Specifically, we use 100 queries each from the HotpotQA and NQ datasets, retrieving top 10 passages from the Wikipedia corpus using the Contriver retriever. We utilize the Wikipedia corpus and retrieval results publicly released by PoisonedRAG (Zou et al., 2024).

Similar to our evaluation with Google Search as the knowledge database, we report the Clean Accuracy (ACC), Robust Accuracy (RACC), and Attack Success Rate (ASR) on the HotpotQA and NQ datasets using the Wikipedia corpus as the underlying knowledge base.

Table 12 reports the clean accuracy across different models, datasets, and values of $\alpha$. The AV Filter preserves high clean performance, with only a modest average drop of $4 - 6\%$ across datasets, with similar trends expected across other configurations.

Table 13 presents the Robust Accuracy (RACC) and Attack Success Rate (ASR) achieved by AV Filter for varying values of $\alpha$ used in computing NormScore$_{\alpha}$. The results demonstrate that the AV Filter often outperforms baseline defenses in robustness, achieving up to $9.8\%$ higher RACC, with similar trends expected across other configurations. Furthermore, even at a low corruption rate of $\epsilon = 0.1$, Vanilla RAG remains highly vulnerable, with ASR reaching up to $90.2\%$. In contrast, AV Filter significantly reduces this vulnerability—bringing the average ASR down to $\mathbf{15.36\%}$ on the HotpotQA and $\mathbf{14.71\%}$ on the NQ dataset.

Table 12: Clean Accuracy (ACC) of defenses, showing that AV Filter preserves RAG utility with a minimal drop from Vanilla, achieving up to $10\%$ higher ACC than other baseline defenses.

| LLM | Mistral-7B | | Llama2-C | | GPT-4o | |
|---|---|---|---|---|---|---|
| **Defense** | **HotpotQA** | **NQ** | **HotpotQA** | **NQ** | **HotpotQA** | **NQ** |
| Vanilla | 51.0 | 59.0 | 36.0 | 46.0 | 45.6 | 47.4 |
| Keyword | **59.0** | 49.0 | **43.0** | 37.0 | 44.6 | **55.0** |
| Decoding | 41.0 | 50.0 | 26.0 | 28.0 | – | – |
| **AV Filter**$_{(\alpha=5)}$ | 40.0 | 43.0 | 27.0 | 34.0 | **45.0** | 48.2 |
| **AV Filter**$_{(\alpha=10)}$ | 46.0 | 44.0 | 27.0 | 36.0 | 44.8 | 47.4 |
| **AV Filter**$_{(\alpha=\infty)}$ | 51.0 | **59.0** | 36.0 | **46.0** | 44.2 | 47.6 |

Table 13: Robust Accuracy and Attack Success Rate (RACC/ASR) showing that AV Filter effectively mitigates attacks with low ASRs while achieving up to $9.8\%$ higher RACC than baselined defenses.

| | Dataset | HotpotQA | | NQ | |
|---|---|---|---|---|---|
| **LLM** | **Attack** | **PIA** | **Poison** | **PIA** | **Poison** |
| | **Defense** | (racc↑ / asr↓) | (racc↑ / asr↓) | (racc↑ / asr↓) | (racc↑ / asr↓) |
| | Vanilla | 18.6 / 69.0 | 14.6 / 75.0 | 22.2 / 55.8 | 23.0 / 50.4 |
| | Keyword | 48.0 / 21.0 | 43.0 / 25.0 | 40.0 / 7.0 | 42.0 / **10.0** |
| **Mistral-7B** | Decoding | 38.0 / 28.0 | 30.0 / 51.0 | 47.0 / 7.0 | 43.0 / 20.0 |
| | **AV Filter**$_{(\alpha=5)}$ | **53.0 / 8.0** | 47.4 / 14.8 | **49.8 / 11.0** | 43.0 / 14.6 |
| | **AV Filter**$_{(\alpha=10)}$ | 52.6 / 8.4 | 47.8 / 15.0 | 48.6 / 11.2 | **44.0** / 13.2 |
| | **AV Filter**$_{(\alpha=\infty)}$ | 47.2 / 13.4 | **48.8 / 13.6** | 36.6 / 26.8 | 42.2 / 12.4 |
| | Vanilla | 3.6 / 90.2 | 14.6 / 65.6 | 6.4 / 85.6 | 26.2 / 48.0 |
| | Keyword | **36.0** / 25.0 | **41.0** / 20.0 | 36.0 / 8.0 | 37.0 / **9.0** |
| **Llama 2-C** | Decoding | 23.0 / 33.0 | 25.0 / **16.0** | 24.0 / 30.0 | 26.0 / 23.0 |
| | **AV Filter**$_{(\alpha=5)}$ | 34.0 / 11.4 | 27.0 / 17.0 | 42.6 / 6.4 | **37.2** / 17.6 |
| | **AV Filter**$_{(\alpha=10)}$ | 34.4 / **10.4** | 26.8 / 17.0 | **44.2 / 6.2** | 36.4 / 15.6 |
| | **AV Filter**$_{(\alpha=\infty)}$ | 17.8 / 44.0 | 21.4 / 28.6 | 22.4 / 45.6 | 32.4 / 25.8 |
| | Vanilla | 10.6 / 78.8 | 20.4 / 58.4 | 16.8 / 69.4 | 28.6 / 34.6 |
| | Keyword | **43.6** / 17.4 | **43.4** / 15.8 | **53.2** / 6.2 | **53.0** / 4.8 |
| **GPT-4o** | **AV Filter**$_{(\alpha=5)}$ | 42.6 / **9.8** | 37.2 / 12.6 | 40.2 / **5.8** | 36.8 / 6.8 |
| | **AV Filter**$_{(\alpha=10)}$ | 40.0 / 11.2 | 37.6 / 12.0 | 41.6 / 6.8 | 35.8 / **4.0** |
| | **AV Filter**$_{(\alpha=\infty)}$ | 35.6 / 17.8 | 41.2 / **11.4** | 28.0 / 29.6 | 38.6 / 5.4 |

Notably, the ASR for the Keyword and Decoding defenses is anomalously high on the HotpotQA dataset. This is attributed to the multi-hop nature of many HotpotQA queries, which often require reasoning across multiple passages. Since both variants of Certified Robust RAG evaluate each passage in isolation, they fail to aggregate information across passages to answer correctly. As a result, they are more susceptible to a single poisoned passage that contains complete information aligned with the adversarial target answer.

### D.7 COMBINING AV FILTER WITH OTHER DEFENSES

As a detection-based pruning defense, AV Filter can be used as a preprocessing step alongside other strategies, such as Certified Robust RAG, to further reduce attack success rates. However, the robust accuracy of the ensemble may still be limited by the underlying defense.

We combine AV Filter ($\alpha = \infty$) with Certified Robust RAG-Keyword by first removing potentially poisoned passages using AV Filter and then applying Keyword for robust generation. Table 14 reports the robust accuracy and attack success rates, with similar trends expected across other configurations. The combined defense consistently achieves lower attack success rates than either method alone, with an average of just **1.22**% across all cases.

Table 14: Robust accuracy and attack success rates for the combined defense. The combination consistently outperforms individual defenses, reducing attack success rates to an average of **1.22%** across all cases.

| Dataset | | RQA-MC | | RQA | | NQ | |
|---|---|---|---|---|---|---|---|
| **LLM** | **Attack** **Defense** | **PIA** (racc↑ / asr↓) | **Poison** (racc↑ / asr↓) | **PIA** (racc↑ / asr↓) | **Poison** (racc↑ / asr↓) | **PIA** (racc↑ / asr↓) | **Poison** (racc↑ / asr↓) |
| **Mistral-7B** | **Keyword** | 57 / 7 | 56 / 6 | 54 / 6 | 55 / 6 | 50 / 1 | **53** / 5 |
| | **AV Filter**$_{(\alpha=\infty)}$ | **79** / 6 | **73** / 8 | **62** / 6 | 54 / 6 | 53 / 7 | 52 / 4 |
| | **Keyword + AV Filter**$_{(\alpha=\infty)}$ | 58 / 3 | 58 / 3 | 60 / 3 | **60** / 3 | **53** / **0** | **53** / **0** |
| **Llama2-C** | **Keyword** | 53 / 6 | 55 / 6 | 53 / 6 | 55 / 6 | 52 / 2 | 53 / 4 |
| | **AV Filter**$_{(\alpha=\infty)}$ | **70** / 18 | **71** / 13 | **61** / 2 | 56 / 6 | **54** / 4 | **54** / 5 |
| | **Keyword + AV Filter**$_{(\alpha=\infty)}$ | 58 / **0** | 58 / **0** | 58 / **0** | **58** / **0** | 53 / 1 | 53 / 1 |
| **GPT-4o** | **Keyword** | 62 / 13 | 60 / 17 | 63 / 15 | **63** / 15 | 58 / 8 | 61 / 6 |
| | **AV Filter**$_{(\alpha=\infty)}$ | 59 / 4 | 50 / 5 | **69** / 2 | 59 / 9 | 59 / 1 | 62 / 1 |
| | **Keyword + AV Filter**$_{(\alpha=\infty)}$ | **64** / 2 | **63** / 2 | 63 / 2 | 63 / 2 | **62** / **0** | **62** / **0** |

## D.8 HYPERPARAMETER ANALYSIS

**Corruption Fraction** $\epsilon$. We evaluate the AV Filter under varying corruption fractions to its robustness as the rate of corruption increases. Specifically, we measure Robust Accuracy (RACC) and Attack Success Rate (ASR) on the RealtimeQA-MC dataset across multiple models, using a fixed $\alpha = \infty$ and a single random seed. Figure 5(a) and (b) present the average RACC and ASR for corruption rates $\epsilon \in \{0.1, 0.2, 0.3, 0.4\}$, with the total retrieved set size fixed at $k = 10$. As expected, increasing the corruption fraction leads to higher ASR and lower RACC. Nevertheless, the AV Filter remains reasonably effective even under high corruption—reducing ASR to 32.67% at $\epsilon = 0.4$ for Poison. We expect a similar trend for other datasets and $\alpha$ values.

**Filtering Threshold** $\delta$. The effectiveness of the AV Filter depends on the filtering threshold $\delta$, which governs the acceptable variance in attention score across the retrieved set. We set $\delta = 26.2$ for our main experiments, estimated from clean retrievals on the RealtimeQA dataset using Llama 2. This estimated threshold generalizes well, as it yields strong performance across different datasets and models. To further assess the robustness of the AV Filter to this hyperparameter, we evaluate its performance across a range of thresholds $\delta \in \{10, 20, 30, 40\}$. Specifically, we report Robust Accuracy (RACC) and Attack Success Rate (ASR) on the RealtimeQA-MC dataset, averaged over multiple models using a fixed $\alpha = \infty$ and a single random seed. Figure 5(c) and (d) show that both RACC and ASR remain relatively stable across this range, indicating that AV Filter is not overly sensitive to $\delta$ and can generalize well to unseen data without requiring fine-tuning. We expect a similar trend for other datasets, attacks, and $\alpha$ values.

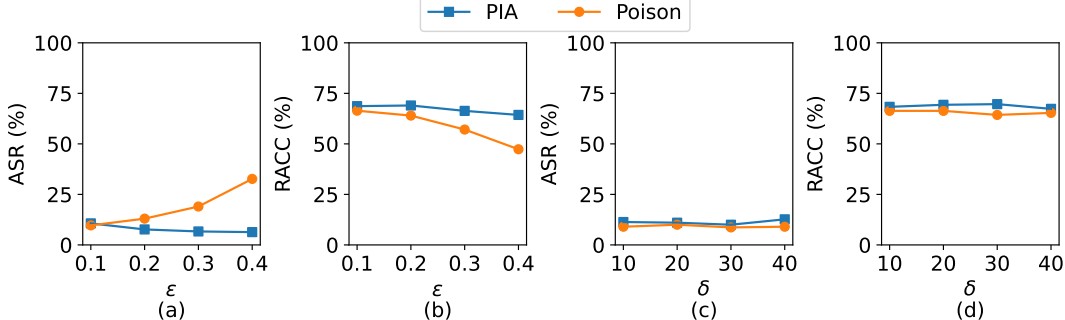

Figure 5: **Effect of Corruption Rate and Filtering Threshold:** This figure shows the impact of varying the corruption rate $\epsilon$ and the filtering threshold $\delta$ on the performance of the AV Filter. Subfigures (a) and (b) present the Attack Success Rate (ASR) and Robust Accuracy (RACC) on the RealtimeQA-MC dataset with $\alpha = \infty$, averaged over all models. As expected, ASR increases and RACC decreases with higher corruption rates. Subfigures (c) and (d) report ASR and RACC for varying $\delta$ values, again averaged over all models, demonstrating that AV Filter's performance is not overly sensitive to the threshold. This indicates that AV Filter can generalize well to unseen data without requiring fine-tuning of $\delta$.

# E  LIMITATIONS

We have shown that existing attacks against RAG systems are not designed for stealth—they often craft poisoned passages that attract anomalously high attention scores, enabling reliable detection and mitigation. This stems from the co-occurrence of the adversary's target answer within the poisoned passage, which causes certain tokens to receive significantly more attention weight than others. When normalized across the retrieved set, these poisoned passages exhibit disproportionately high attention scores, resulting in a high-variance signal that AV Filter leverages for detection.

However, this detection strategy assumes that the poisoned content is concentrated in a small subset of passages while the majority support the correct answer. This reliance leads to certain limitations. To the best of our knowledge, using attention patterns to improve the robustness of RAG systems has the following constraints:

1. **Susceptibility to majority corruption.** If the adversary manages to corrupt a majority of the retrieved set, then multiple passages will contain tokens that draw high attention weight. This the contrast of normalized attention scores among passages, reducing the variance and potentially allowing the attack to evade detection by AV Filter. This highlights the need for robustness at the retrieval stage (*Step I*) of the RAG pipeline as well. However, AV Filter's improvements are orthogonal to retrieval robustness—it can be integrated with more robust retrievers that make majority corruption harder.

2. **Dependence on redundancy of correct knowledge.** If only a very few benign passages contain the correct answer, these may individually attract high attention and be mistakenly filtered out. Thus, AV Filter assumes that the knowledge corpus includes multiple passages supporting the correct answer, which is necessary for any filtering mechanism based on outlier detection to succeed.

3. **Task specificity and generalization.** The AV Filter relies on the poisoned passage receiving high normalized attention scores due to the co-occurrence of the target response. While this is well-suited for question-answering tasks—where the goal is to inject or alter the response—we have not yet evaluated its effectiveness against attacks that aim to exploit other behaviors of the RAG system (e.g., manipulating style, eliciting private data, or controlling downstream decisions) without directly changing the response content. Broader evaluations will be necessary to understand the generalization of this defense mechanism.

## F  LLM USAGE

We did not make use of LLMs in the writing or research process beyond minor revisions to the text.

## G  BROADER IMPACTS

We propose a filtering technique capable of identifying and mitigating existing poisoning attacks, thereby reducing potential harm. In parallel, we introduce more stealthy poisoning attacks that evade existing defenses. While we believe this dual contribution will drive the development of more robust RAG systems, it may also increase the risk to vulnerable deployments in the short term.

