# OpenReview forum: "Through the Stealth Lens: Attention-Aware Defenses Against Poisoning in RAG"
_ICLR.cc/2026/Conference — Submitted to ICLR 2026_

### Official Review · Reviewer_JgSt · 2025-10-30

**Soundness:** 3
**Presentation:** 3
**Contribution:** 2
**Rating:** 4
**Confidence:** 3

**Summary:**

This paper proposes the Attention-Variance Filter (AV Filter), a novel defense against poisoning attacks in Retrieval-Augmented Generation (RAG) systems. The core idea is that malicious passages, in order to successfully poison the output, must attract a disproportionate amount of the model's attention. The authors formalize this using a Normalized Passage Attention Score (NPAS) and show that the variance of these scores across passages is a reliable signal for detecting corruption. The proposed AV Filter iteratively removes passages with the highest attention scores until this variance drops below a threshold. The paper evaluates this defense against several content-poisoning and instruction-poisoning attacks, demonstrating its effectiveness in reducing the Attack Success Rate (ASR) while claiming to maintain high accuracy in benign settings.

**Strengths:**

1. Novel Defense Signal: The use of attention variance as a signal for detecting poisoned context is novel.

2. Thorough Experimental Setup: The authors conduct a comprehensive set of experiments across multiple datasets (RQA, NQ, HotpotQA), LLMs (Llama2, Mistral, Llama-3.1, Deepseek-R1, GPT-4o), and attack types (Poison, PIA, MA, Paradox).

3. Baseline Comparison: The paper includes a comparison against the state-of-the-art "Certified Robust RAG" (Keyword and Decoding) baselines, providing a clear benchmark for its performance.

**Weaknesses:**

1. Weak Reranker Baseline: The comparison to reranking methods in Appendix D.5 (Table 11) may have some problems. The paper's baseline "removes the passage ranked highest in relevance," which is the opposite of a reranker's intended use. A proper baseline would use the reranker to re-order the passages and pass the most relevant ones to the LLM. Furthermore, the paper uses relatively weak rerankers (ColBERTv2, T5). Even against this  baseline, Table 11 shows that for GPT-4o, the reranking method outperforms AV Filter in 4 out of 9 setups. A comparison against modern, stronger rerankers  using a correct methodology is necessary to prove the utility of AV Filter.

2. Inconsistent Performance Against "Keyword" Baseline: The "Keyword" baseline from Xiang et al. (2024) proves to be a very strong competitor. As seen in Table 2, the Keyword baseline delivers significantly better Robust Accuracy (RACC) and lower Attack Success Rate (ASR) than AV Filter in several key scenarios, particularly against Prompt Injection Attacks (PIA) on stronger models like Deepseek-R1 and GPT-4o. This makes the practical necessity of the more complex AV Filter unclear.

3. Remaining Accuracy Drop: In many other cases, the AV Filter does result in a drop in clean accuracy (e.g., an average of 4-6% noted by the authors). While this may be an acceptable trade-off for some, when combined with the inconsistent performance against baselines, it weakens the overall proposal.

**Questions:**

1. Explanation of Baselines: The "Keyword" and "Decoding" baselines are central to your evaluation, how do they work? what are the specific configurations for these two methods?

---

> ### Author Response · Authors · 2025-11-19
> **Response to Reviewer JgSt (1/2)**
>
> We thank the reviewer for their detailed feedback. We address their comments in detail below:
>
> ## 1. Comparison with the Baselines.
> We use Certified Robust RAG as our main baseline. We explain the design of Cert. Robust RAG in detail, some of which is already included in Appendix A. The design of Certified Robust RAG is inherently limited when retrieval across multiple passages is required, and, in the case of Keyword, when detailed contextual information is needed from any single passage.
>
> Certified Robust RAG employs an isolate-then-aggregate strategy. Given $k$ retrieved passages, it first computes an individual LLM response for each passage in isolation, rather than concatenating all $k$ passages as in a standard RAG pipeline. It then aggregates these responses using two different methods, Keyword and Decoding, to produce the final answer.
>
> **Keyword.** It uses the $k$ isolated responses from the retrieved passages. Among these responses, it discards those whose output is equivalent to “I don’t know,” leaving $k’$ passages. It then extracts unique keywords from these $k’$ responses, and a keyword is retained if its count exceeds min($\alpha, \beta \cdot k’$). For a corruption of 1 in 10, it uses $\alpha=3$ and $\beta=0.3$. In many cases, $k’$ is very small because many passages respond with “I don’t know”. As a result, $0.3 \cdot k’ < 1$, which means that all extracted keywords from the $k’$ responses are effectively accepted. Finally, these keywords are used, rather than the original passages, to query for the final response.
>
> **Decoding.** It uses each passage in isolation to generate a next-token probability distribution. This produces $k$ next-token distributions, which are then aggregated by taking their mean. The final next token is drawn from this averaged distribution to produce a robust response.
>
> This isolation technique, combined with the aggressive removal of context used by Keyword, makes successful attacks more difficult but also substantially reduces utility. **In multi-hop settings like HotpotQA, where answering requires jointly reasoning over multiple documents, such a design is fundamentally incompatible: once documents are isolated, the LLM will almost respond “I don’t know”**. Consistent with this, Table 13 in the Appendix shows that **AV Filter outperforms Keyword on HotpotQA in all 6/6 evaluation settings, yielding better ASR.**
>
> That said, we acknowledge that there are some settings where Certified RAG outperforms our method. However, its highly restrictive design necessarily makes it unusable in practical scenarios faced by real-world RAG systems. As more realistic and comprehensive benchmarks emerge, we expect these limitations to become even clearer.
>
> >This makes the practical necessity of the more complex AV Filter unclear.
>
> AV Filter offers a strong design advantage due to its simplicity and low false-positive rate. Unlike certified RAG approaches, it does not alter the inference pipeline; it merely filters out potentially poisoned passages. This modularity allows AV Filter to be combined seamlessly with other defenses. **As shown in Appendix D.7, pairing AV Filter with Keyword yields consistent gains: AV Filter + Keyword outperforms Keyword alone in all 18/18 settings, improving both robust accuracy and ASR.**
>
> &nbsp;
> ## 2. Drop in Clean Accuracy.
>
> As the reviewer correctly noted, AV Filter introduces a 4-6% average drop in clean accuracy. Importantly, this degradation is not unique to AV Filter, any robust defense that restricts or filters the RAG outputs will experience some clean accuracy loss unless it can perfectly distinguish every clean passage from every poisoned one.
>
> **For comparison, the average clean accuracy drops for other mechanisms are even larger: Keyword incurs a 7.16% drop, and Decoding results in an 11.81% drop.**

---

> ### Author Response · Authors · 2025-11-19
> **Response to Reviewer JgSt (2/2)**
>
> ## 3. Reranker Baseline.
> As the reviewer noted, the typical purpose of a reranker is to reorder retrieved passages so that the most relevant ones are passed to the LLM. However, this relevance-based ordering is not robust to adversarial corruption. **In our study, we consistently observed that reranking models assign very high relevance scores to poisoned passages because these passages do contain a clear answer to the query; the answer is simply incorrect.** Since standard rerankers are not trained to anticipate or detect adversarial corruption, they naturally treat poisoned passages as highly relevant.
>
> This is why, in our experiments, we removed the passage with the highest relevance score under the assumption that it might be poisoned. Yet even this strategy is not robust: an adaptive attacker can easily manipulate the poisoned passage to become the second-most relevant while still containing the malicious content.
>
> **This discrepancy between relevance scoring and the actual trustworthiness of passages highlights that rerankers, by themselves, are insufficient for mitigating adversarial corruption.** Moreover, this reordering is not tied to the actual inference mechanism, unlike our technique which introduces a real tradeoff between stealth and attack success rate.
>
> >A comparison against modern, stronger rerankers using a correct methodology is necessary to prove the utility of AV Filter.
>
> As suggested by the reviewer, we now report the ASR when providing the top $(1 - \epsilon) \cdot k$ relevant passages according to Cohere Rerank-3.5 below:
>
> ### Attack Success Rates for Reranking Baseline (Cohere Rerank-3.5)
> | **LLM** | **Defense** | **RQA-MC: PIA** | **RQA-MC: Poison** | **RQA: PIA** | **RQA: Poison** | **NQ: PIA** | **NQ: Poison** |
> |:--------:|:-------------:|:------------------:|:----------------------:|:--------------:|:-------------------:|:--------------:|:------------------:|
> | **Mistral-7B** | Reranking (Cohere) | 33 | 26 | 18 | 14 | 11 | 5.0 |
> | | AV Filter ($\alpha = \infty$) | 7.2 | 8.4 | 2.4 | 8.6 | 5.8 | 4.0 |
> | **Llama2-C** | Reranking (Cohere)| 64 | 35 | 93 | 16 | 76 | 11 |
> | | AV Filter ($\alpha = \infty$) | 16.8 | 12.6 | 5.0 | 6.60 | 9.2 | 3.6 |
> | **GPT-4o** | Reranking (Cohere) | 9 | 13 | 21 | 26 | 21 | 3.0 |
> | | AV Filter ($\alpha = \infty$) | 5.4 | 6.8 | 7.0 | 9.2 | 11.4 | 1.6 |
>
> &nbsp;
>
> **We thank the reviewer for their feedback. In light of these explanations, we kindly request the reviewer to consider increasing the rating. We would be happy to provide any further clarifications if needed.**

---

> ### Author Response · Authors · 2025-11-25
> **Request for Reviewer Feedback on Addressed Concerns**
>
> We wanted to follow up and check whether any further clarification is needed from our side. We believe we have addressed the earlier concerns and would appreciate your response when convenient.

---

> ### Author Response · Authors · 2025-11-28
> **Summary of Rebuttal**
>
> We provide below a brief summary of our clarifications; please refer to our previous responses for detailed explanations of each point.
>
> **1. Comparison with Baselines.** Certified Robust RAG's isolate-then-aggregate design limits utility, especially for multi-hop tasks (e.g., HotpotQA, Table 13). AV Filter is modular, low-overhead, and even improves certified RAG in all 18/18 cases without modifying the inference pipeline. _Please refer to our previous response (1/2) for detail._
>
> **2. Drop in Clean Accuracy.**  The 4-6% clean accuracy drop is an expected tradeoff common to all robust defenses and remains smaller than Keyword (7.16%) and Decoding (11.81%), demonstrating a better robustness-utility balance. _Please refer to our previous response (1/2) for detail._
>
> **3. Reranker Baseline.** Rerankers frequently assign very high relevance to poisoned passages, making reranking inherently non-robust. As the reviewer suggested, we evaluated the typical reranker baseline using Cohere Rerank-3.5 (removing the least relevant passage), and AV Filter yields consistently lower ASR across datasets and models. _Please refer to our previous response (2/2) for detail._
>
> **We believe our responses address the reviewer’s concerns and would be happy to provide further clarification if needed.**

---

### Official Review · Reviewer_WDZX · 2025-10-30

**Soundness:** 4
**Presentation:** 4
**Contribution:** 4
**Rating:** 8
**Confidence:** 5

**Summary:**

The paper introduces a Stealth Attack Distinguishability Game (SADG) to formally characterize the trade-off between stealthiness and detectability of poisoning attacks against Retrieval-Augmented Generation systems. To analyze this relationship, the authors leverage attention signals to quantify the influence of each retrieved passage on the model’s final answer. Building on this observation, they design a lightweight anomaly detector called the AV Filter, which identifies and removes suspicious passages based on abnormal variance in normalized passage-level attention scores. Experiments demonstrate that the AV Filter significantly reduces attack success rates across multiple datasets and language models, while maintaining high clean performance.

**Strengths:**

1. The paper analyzes the influence of poisoned passages through generation-time attention signals, providing an intuitive and interpretable view of which retrieved segments drive the model’s responses. Building on this insight, the authors design a lightweight and effective defense algorithm, making the motivation both clear and convincing.
2. By proposing the Stealth Attack Distinguishability Game (SADG), the paper introduces a quantitative framework that defines “stealth” as a game-theoretic metric. This formalization contributes a principled foundation for future theoretical and adversarial analyses of RAG security.
3. The experiments are thorough, covering multiple datasets, various large language models, and different types of poisoning attacks, including both content-based and instruction-based threats. Moreover, the authors design a novel adaptive attack to stress-test their defense, further demonstrating the robustness of the evaluation.

**Weaknesses:**

1. The proposed defense assumes that only a small fraction of retrieved passages are poisoned. If an attacker manages to contaminate the majority of retrieved entries, especially in private or small-scale knowledge bases, the variance signal used by the AV Filter can be diluted, leading to failure of detection. This limitation poses a realistic concern for practical deployments.
2. The AV Filter is primarily designed to defend against attacks that alter the content of generated answers in QA-style RAG tasks. Its effectiveness in other non-QA or generative contexts, such as summarization, dialogue, or decision-making tasks, remains untested.
3. The paper introduces numerous formulas and symbols that are not strictly necessary for understanding the core ideas. This abundance of notation may reduce the overall readability.

**Questions:**

Please refer to my comments on weaknesses.

---

> ### Author Response · Authors · 2025-11-17
> **Response to Reviewer WDZX**
>
> We thank the reviewer for their positive feedback and for appreciating our work. We address their comments in detail below:
>
> ## 1. Fraction of corruption $\epsilon < 0.5$ in the retrieved set.
> As the reviewer correctly highlighted, we assume that the adversary cannot corrupt the majority of the retrieved set, since robustness becomes theoretically impossible for any generation-step defense under such conditions.
>
> If an adversary manages to outnumber the benign documents in a small-scale knowledge base for a given query, i.e., the poisoned documents dominate, then it again becomes information theoretically impossible to provide robust performance from RAG.
>
> These limitations have motivated two main goals for our future work:
>
> 1. **Develop a robust retriever** that manages to return as many benign documents as possible when the knowledge base has a majority of benign content for a query. This may require introducing non-trivial randomization into the retrieval mechanism, significantly increasing the difficulty of optimizing the adversary's objective while still maintaining utility.
>
> 2. **Develop a secure method for adding new documents to the knowledge base.** Starting from a verified small base, our goal is to incorporate new content efficiently while limiting extreme statistical shifts in the knowledge base relative to typical queries.
>
> Both goals require thorough scientific investigation and are part of our planned future work.
>
>
> ## 2. Design for other non-QA tasks.
> We currently consider only definitive QA tasks with fixed ground-truth responses, similar to most related works. We mention this as a limitation of our current study, but not as a definitive limitation of the AV Filter technique itself. Extending AV Filter to other task types is an interesting and important direction, which we view as part of our future work.
>
> At present, AV Filter uses attention weights from all tokens in a response, assuming that each token in a QA response contributes meaningful information and can exhibit strong correlation with the passage supporting that token. This assumption may not hold for responses in other types of tasks. Our next aim is to **adaptively select tokens** from responses for computing our attention score, yielding a better estimate of correlation and making the approach even harder for an adversary to optimize against.
>
> &nbsp;
>
> **We thank the reviewer again for their thoughtful feedback. We will also revise the manuscript as suggested to improve readability. We kindly request the reviewer to show strong support and champion our work.**

---

### Official Review · Reviewer_oHcu · 2025-11-03

**Soundness:** 3
**Presentation:** 3
**Contribution:** 3
**Rating:** 2
**Confidence:** 5

**Summary:**

The authors propose an attention-aware defense mechanism designed to mitigate the impact of knowledge attacks on Retrieval-Augmented Generation systems.

**Strengths:**

1. The paper is clearly written and straightforward to understand.

2. The authors perform experiments to demonstrate the effectiveness of the proposed defense method.

**Weaknesses:**

1. The threat model assumed in the paper is strong.

2. The proposed approach may fail when $\epsilon$ exceeds 0.5.

3. Several existing attack and defense methods are not included in the evaluation.

**Questions:**

1. The paper assumes that the defender has full access to the model’s internal attention matrices and can compute multi-layer attention scores. This assumption is impractical for many real-world RAG systems that rely on closed-source APIs (e.g., GPT-4, Claude). Although the authors mention experiments with a “closed-source” setup, the proposed defense still depends on internal attention data, which limits its applicability in practical deployments.

2. In the threat model, the authors assume that the attacker can inject only a few poisoned passages into the top-k retrieval results, with $\epsilon < 0.5$. This is a strong assumption. Prior works [a][b] generally assume that the attacker can poison only a small portion of the overall knowledge database. For example, in [a], for each targeted query, five poisoned passages are injected while considering the top-5 retrieval results, meaning that the fraction of poisoned passages $\epsilon$ can exceed 0.5 or even reach 1.0 (all retrieved passages are poisoned).

3. In the experiments, the authors evaluate the defense under a mild adversarial setting where the corruption rate $\epsilon = 0.1$ and top-10 results are retrieved. This setting is too weak. Following the previous comment and the setups in [a][b], it would be more convincing to report results when five poisoned passages are injected per query and top-5 results are retrieved in the RAG framework.

4. Several recent attacks specifically targeting RAG systems have been proposed, such as [b][c][d]. The authors should also evaluate the robustness of their defense against these more advanced attacks.

5. Numerous new defenses have also been developed. The authors should include comparisons with these recent defense methods, such as [e][f], to provide a more comprehensive evaluation.

6. As discussed earlier, the corruption rate $\epsilon$ can exceed 0.5 in top-k retrieval results. Therefore, for Figure 5, it would be helpful to also include experimental results for larger $\epsilon$ values, such as 0.7, 0.9, or 1.0.



[a] PoisonedRAG Knowledge Poisoning Attacks to Retrieval-Augmented Generation of Large Language Models.

[b] Practical Poisoning Attacks against Retrieval-Augmented Generation.

[c] FlippedRAG Black-Box Opinion Manipulation Adversarial Attacks to Retrieval-Augmented Generation Models.

[d] Machine Against the RAG Jamming Retrieval-Augmented Generation with Blocker Documents.

[e] Certifiably Robust RAG against Retrieval Corruption.

[f] Traceback of Poisoning Attacks to Retrieval-Augmented Generation.

---

> ### Author Response · Authors · 2025-11-16
> **Response to Reviewer oHcu (1/2)**
>
> We thank the reviewer for their detailed feedback and for referencing related works. We clarify the questions in detail below and provide additional experimental results.
>
> ## 1. Access to LLM’s internal attention weights for defense.
> Access to attention matrices is fully justified when the defender is the model provider. Furthermore, using them as corruption signals adds minimal overhead since these matrices are already computed during inference. This low overhead also makes the approach a viable option for consideration by "closed-source" API models in their robust deployments.
>
> We acknowledge that many real-world RAG systems rely on closed-source API models. In such cases, our technique can be applied using a surrogate local LLM, whose attention patterns help filter potentially poisoned passages with minimal overhead. This does not limit practical applicability; our experiments show that surrogate attention consistently improves robustness even for closed-source models like GPT-4o. **Table 2 reports the attack success rate of GPT-4o when AV Filter is applied using Mistral-7B (randomly chosen) as a surrogate model. On average, the attack success rate drops significantly from 23.8% in Vanilla RAG to 4.93% with AV Filter.** This demonstrates that AV Filter effectively strengthens closed-source models as well for real-world RAG deployments.
>
> **Even the prior work [c] cited by the reviewer uses surrogate models to craft attacks against RAG systems**, so defenders should likewise be allowed to use local LLMs for defense. More broadly, having access to the deployed system is a key advantage for any defender; expecting strong protection in a strictly closed-source setting without additional resources is inherently limiting.
>
> In particular, this restriction rules out a promising line of defenses using internal behaviors of the model, such as attention, for mitigating diverse LLM attacks. Recent work shows its effectiveness in countering threats such as jailbreaks [g] and prompt-injection attacks [h].
>
> [g] Universal Jailbreak Suffixes Are Strong Attention Hijackers
>
> [h] May I have your Attention? Breaking Fine-Tuning based Prompt Injection Defenses using Architecture-Aware Attacks
>
> ## 2. Assuming fraction of corruption $\epsilon < 0.5$ in the retrieved set.
> **Multiple related works, including [b] and [e] mentioned by the reviewer, support our assumption.**
>
> Our work targets robustness in the generation step of a RAG pipeline, which is orthogonal to improving retriever robustness. Since perfectly robust retrieval is unrealistic, the generator should tolerate minimal corruption so it can operate reliably with a reasonably robust retriever (e.g., Google Search). Designing robust generation mechanisms that withstand sub-majority corruption ($\epsilon < 0.5$) sets a far more attainable goal for any practical robust retriever.
>
> If an attacker corrupts more than half of the retrieved set, no generation-step defense can remain robust. Thus, we operate under the standard assumption $\epsilon < 0.5$, consistent with related work [e][b]. This also underscores the importance of developing more robust retrievers, an essential direction for achieving true end-to-end robustness and a focus of our future work.
>
> Even with a perfectly robust RAG pipeline, we must still assume the attacker cannot outnumber benign documents in a knowledge base for a given query; once poisoned passages dominate, robustness becomes theoretically impossible.
>
> **CorruptRAG [b] injects only a single poisoned passage per query** and argues that outnumbering benign passages is unrealistic in practice, as doing so is costly, resource-intensive, and increases the likelihood of detection. It argues that PoisonedRAG [a] assumes a very strong adversarial setting where the system is dominated by poisoned rather than reliable information. The paper explicitly states:
> > only few queries have 4 texts as ground-truth relevant, with most queries containing fewer than 3 ground-truth texts among the top-5. In contrast, the PoisonedRAG method inserts 5 poisoned texts per query into the knowledge database, ensuring that the number of poisoned texts surpasses the ground-truth texts. This shows that PoisonedRAG not only proves costly but also impractical, as it would cause the system to become dominated by poisoned rather than reliable information.
>
> Moreover, both PoisonedRAG and CorruptRAG achieve high ASR even with a single poisoned passage per query. Such minimal corruption is difficult to detect even for strong retrievers, further underscoring the need to provide robustness under this setup before moving to stronger adversarial settings.
>
> **We provide our main results in Table 2 with $\epsilon = 0.1$, consistent with the baseline [e] mentioned by the reviewer. We argue that this setting is not overly weak, as the adversary still achieves high ASR against Vanilla RAG under this budget.**

---

> ### Author Response · Authors · 2025-11-16
> **Response to Reviewer oHcu (2/2)**
>
> ## 3. Attack evaluation.
>
> We evaluate our defense against four attacks, including PoisonedRAG, Prompt Injection, Misinformation Attack, and RAG Paradox, covering both content-based and instruction-based threats.
>
> **We further stress-test the defense using a strong adaptive attack with full knowledge of both our defense and the entire RAG pipeline, as described in Appendix D.1.** In this adaptive attack, we use GCG and AutoDAN to craft poisoned passages. We argue that surpassing the strength of our adaptive attack would require significantly more advanced optimization methods as well as retaining full access to the system.
>
> **We also include the CorruptRAG-AS [b] attack, as suggested by the reviewer**, and report its results below for $\epsilon=0.2$, using the same hyperparameters as in Table 2.
>
> ### CorruptRAG-AS (Robust Acc. / ASR)
> | Model       | Defense / Dataset  | RQA-MC        | RQA           | NQ            |
> |---|---|--|--|-|
> | **Mistral 7B** | Vanilla  | 52.0 / 32.0 | 43.6 / 17.8  | 40.4 / 15.8  |
> |             | AV-Filter α=5  | 69.4 / 10.8 | 57.2 / 7.2    | 48.4 / 9.2    |
> |             | AV-Filter α=10| 68.2 / 11     | 59.2 / 8.0  | 51.0 / 9.2    |
> |             | AV-Filter α=∞ | 74.6 / 7.6    | 61.8 / 7.4  | 51.2 / 8  |
> | **Llama 2** | Vanilla      | 37.0 / 51.8     | 20.0 / 63.4   | 44.2 / 30.6   |
> |             | AV-Filter α=5       | 45.0 / 39.8      | 54.0 / 15.4  | 42.8 / 19.8   |
> |             | AV-Filter α=10      | 49.0 / 34.8    | 53.0 / 17.4   | 45.6 / 18   |
> |             | AV-Filter α=∞       | 52.6 / 18.8     | 45.2 / 38.8  | 46.2 / 22.4    |
> | **GPT-4o**  | Vanilla             | 53.4 / 4.6    | 61.2 / 11.6    | 56.8 / 2.4   |
> |             | AV-Filter α=5       | 54.2 / 3.6    | 66.0 / 6.8   | 56.8 / 6.0     |
> |             | AV-Filter α=10      | 54.8 / 3.2       | 65.6 / 6.0     | 57.0 / 5.4    |
> |             | AV-Filter α=∞       | 55.2 / 3.0     | 65.2 / 5.2    | 59.8 / 3.4    |
> | **Llama 3.1**  | Vanilla       | 22.0 / 13.8   | 35.8 / 24.0   | 31.8 / 25.0   |
> |             | AV-Filter α=5       | 25.4 / 9.2    | 64.0 / 5.4   | 54.2 / 7.6    |
> |             | AV-Filter α=10      |27.8 / 8.2       | 63.6 / 3.6     | 54.8 / 6.2   |
> |             | AV-Filter α=∞       | 31.4 / 6.8     | 65.6 / 2.8   | 55.4 / 8.0    |
> | **Deepseek-R1**  | Vanilla | 35.0 / 7.4   | 35.4 / 14.2    | 38.2 / 6.0  |
> |             | AV-Filter α=5       | 33.4 / 5.4   | 40.4 / 8.4   | 43.6 / 4.8     |
> |             | AV-Filter α=10      | 33.0 / 6.0      | 40.2 / 8.8     | 41.4 / 4.4   |
> |             | AV-Filter α=∞       | 33.8 / 6.2    | 35.8 / 12.2   | 37.6 / 7.4    |
>
> The other attack FlippedRAG [c] mentioned by the reviewer aims only to optimize retrieval corruption, and the actual poisoned passage (targeted opinion) responsible for the malicious response is taken from a fixed, task-specific dataset. Because it provides no mechanism for generating new poisoned passages, it cannot be adapted to our tasks.
>
> Similarly, the Jamming attack [d] mentioned by the reviewer is effectively a subset of PoisonedRAG and Prompt Injection, and thus is expected to yield similar results.
>
> ## 4. Defense evaluation.
>
> We compare our defense to the state-of-the-art Certified Robust RAG [e] defense and report the results in Table 2. We also compare our defense with other baseline defenses, including Perplexity Filtering, Vigilant Prompting, and Reranking Methods, in Appendix D.5.
>
> > The authors should include comparisons with these recent defense methods, such as [e][f].
>
> **We already compare our defense with [e] as our main baseline and report the results in Table 2 of the main text.**
>
> RAG Forensics [f], mentioned by the reviewer, is not technically a defense. It attempts to trace back the poisoned passage with the help of feedback from the user (confirming that the response is incorrect). It simply queries an LLM for a label indicating whether a particular passage is poisoned given each passage and the response. If we attempt to adapt this into a defense, it essentially reduces to asking an LLM which passage is corrupted. This approach is not robust, as a poisoned passage can coerce the LLM into always classifying it as clean. Moreover, this classification is not tied to the actual inference mechanism, unlike our technique.
> &nbsp;
>
> &nbsp;
>
> **Overall, robust defenses against RAG poisoning are still scarce. While no defense, including ours, covers every scenario, our approach represents a meaningful step toward building end-to-end robust RAG pipelines.**
>
> **We thank the reviewer for their feedback. We will include all the additional related works in our revised manuscript. In light of these explanations, we kindly request the reviewer to consider increasing the rating. We would be happy to provide any further clarifications if needed.**

---

> ### Author Response · Authors · 2025-11-25
> **Request for Reviewer Feedback on Addressed Concerns**
>
> We wanted to follow up and check whether any further clarification is needed from our side. We believe we have addressed the earlier concerns and would appreciate your response when convenient.

---

> > ### Comment · Reviewer_oHcu · 2025-11-27
> >
> > All my concerns have been addressed, so I have decided to increase my score.

---

> > > ### Author Response · Authors · 2025-11-27
> > >
> > > We once again thank the reviewer for their detailed feedback and for increasing their rating.

---

### Official Review · Reviewer_sWjW · 2025-11-07

**Soundness:** 3
**Presentation:** 3
**Contribution:** 3
**Rating:** 6
**Confidence:** 3

**Summary:**

The paper investigates defenses against data poisoning attacks in retrieval-augmented generation (RAG) systems. The key observation is that existing poisoning attacks tend to lack stealth: tokens in the target responses specified by such attacks exhibit disproportionately high correlations with the poisoned input passages. Building on this observation, the paper introduces the Normalized Passage Attention Score (NPAS), which measures each retrieved passage’s contribution to the response based on the attention weights within the LLM component of the RAG system. By analyzing the variance of NPAS across retrieved passages, they formulate a Stealth Attack Distinguishability Game (SDAG) that identifies potentially corrupted or poisoned passages and further proposes an attention-variance filter to filter out those corrupted passages. Experiments demonstrate the effectiveness of the proposed defense against the state-of-the-art defense baseline across five attacks on four QA benchmark datasets.

**Strengths:**

1. The paper is well-presented overall, with the problem setup and attack model clearly defined and formalized, which significantly aids reader comprehension of both the context and the proposed defense.

2. Developing defenses against data poisoning attacks in retrieval-augmented generation (RAG) is a problem of high practical relevance.

3. While the general idea of leveraging attention scores to detect correlations between poisoned inputs and target outputs as a defense method is not entirely new, its adaptation to the RAG defense setting could potentially be interesting.

**Weaknesses:**

1. Several aspects of the proposed defense method described in Section 4 require further clarification (see the specific questions below).

2. The experiment results in terms of attack success rate (ASR) do not seem to be strong. In particular, Table 2 compares the proposed defense with the state-of-the-art baseline, Certified Robust RAG [Xiang et al., 2024], but the results indicate that the proposed method yields higher ASR values in most settings, suggesting weaker defense effectiveness. For example, when defending against the Poison attack, across five models evaluated on three datasets, the proposed defense achieves an equal ASR compared to the baseline in only one case.

**Questions:**

Regarding Section 4 on the proposed defense:

1. In the subsection “Discriminating Bbetween Corrupted and Benign Retrievals via Attention”, how are the two sets of retrieved passages constructed? Is the assumption that one set contains only benign passages while the other contains corrupted ones? What if both sets are corrupted?

2. In lines 312-313, the paper assumes that “the attention distribution over passages is approximately uniform with a slight recency effect.” How valid is this assumption for extractive QA queries, where attention typically concentrates on the passage region containing the correct answer? Would such cases undermine the proposed discriminative mechanism?

3. In line 332, it is stated that sorting passages by attention scores helps reduce the recency effect. Could the authors clarify why and how this sorting step mitigates the effect?

4. How to choose the $\alpha$ threshold to compute NPAS in practice?

------

Other questions:

5. In Table 1, the Clean Accuracy of the proposed method with Deepseek-R1 and Llama-3.1 exceeds that of the Vanilla RAG on the RQA and NQ datasets, respectively. Would it be possible to explain why this is the case here?

---

> ### Author Response · Authors · 2025-11-14
> **Response to Reviewer sWjW (1/2)**
>
> We appreciate the positive feedback and support for our work. Below, we address the questions and concerns in more detail, and we will incorporate these clarifications into the revised manuscript as well.
>
> ## 1. Lower ASR for Certified RAG.
> We acknowledge that Cert. RAG Keyword exhibits a very strong Attack Success Rate (ASR). It outperforms AV Filter 23/30 times in Table 2. The average ASR of Keyword is 3.97 as compared to 5.84 (for the best α in each case). Keyword achieves this strong ASR because of its overly strict approach to pruning passages and selecting the final answer, which is reflected in its inability to provide the robust answer (correct response), resulting in lower robust accuracy in 17/30 cases.
>
> Certified RAG Keyword employs an isolate-then-aggregate strategy. Given k retrieved passages, it computes k individual responses, one from each passage in isolation. It extracts the unique keywords from each response and maintains a count for each keyword over all responses. It then selects only those keywords whose count exceeds the number of possibly corrupted passages and uses these keywords to generate the final answer. If no such keyword exists, it uses all keywords for generating the final answer. In cases where no keyword appears in any response (for example, when all responses are “I don’t know”), it discards all passages and just uses the query.
>
> This isolation technique and aggressive removal of context make it hard to produce a successful attack but significantly reduce utility. Consider the HotpotQA task, where questions require finding and reasoning over multiple supporting documents; such a design will never work, as the individual responses will always be “I don’t know.” **Table 13 in the Appendix shows the results on the HotpotQA dataset, where AV Filter achieves better ASR than Keyword in all 6/6 cases**.
>
> AV Filter also holds a strong advantage in terms of its design. It is a simple filtering mechanism that removes potentially poisoned passages without changing the inference pipeline, unlike certified RAG. This allows us to combine AV Filter with any other defense. **Appendix D.7 shows the results of combining AV Filter with Keyword: AV Filter + Keyword outperforms Keyword in all 18/18 cases, both in terms of robust accuracy and ASR.**
>
> Thus, we argue that a filtering mechanism like AV Filter, which uses the variance of attention scores as a signal of corruption, is useful because it will continue to improve with better techniques for estimating attention influence, is effective as a stand-alone defense, and can enhance other defenses as well due to its low false-positive rate.
>
> ## 2. Discriminating Between Corrupted and Benign Retrievals via Attention.
> This subsection highlights how the variance of attention scores across passages serves as an effective signal for distinguishing between benign retrieval sets and corrupted retrieval sets. We obtain a benign retrieved set for each question (i.e., the top-10 Google search results) and then construct a corrupted retrieved set by randomly replacing one passage with a poisoned passage. Figure 2(b) shows how the distribution of variance in the corrupted retrieval set differs significantly from that of the benign one. This demonstrates the motivation for using attention variance as a corruption signal and removing passages until the variance drops below a certain threshold as a defense mechanism.
>
> ## 3. Attention distribution over passages is approximately uniform.
> Lines 312–313 describe that the distribution of attention scores (NPAS) in the benign retrieved set is approximately uniform across passages, meaning each passage receives roughly 10% NPAS when k = 10, as shown in Figure 2(a). There is a slight recency effect in which passages closer to the response (toward the end of the prompt) receive slightly more attention. This comment was not related to how attention weights are distributed within a single passage. We discuss that more explicitly in Appendix C. **We will update the sentence to state “distribution of NPAS” to make this clearer and avoid this confusion.**

---

> ### Author Response · Authors · 2025-11-14
> **Response to Reviewer sWjW (2/2)**
>
> ## 4. How does sorting help to overcome the recency bias?
> **Sorting aims to make filtering independent of the order in which passages were returned by the retriever.** As mentioned earlier and in the manuscript, attention patterns in LLMs exhibit a recency bias. This can create an issue when attempting to use NPAS as a corruption signal. Consider an example of a corrupted retrieved set with NPAS scores [7,15,9,7,8,10,9,9,10,16], where index 1 is the poisoned passage. Even though index 1 has an anomalously high attention score for its position, it is still lower than the score of the last passage due to the recency effect. Simply removing the passage with the highest NPAS would therefore not be robust in this case.
>
> Our solution is to sort the passages once according to NPAS. In this example, the poisoned passage would move to index 8. When calculating NPAS again for this sorted order, the poisoned passage now receives very high attention due to both its strong malicious influence on the incorrect response and the recency effect, making it a clear outlier in the NPAS distribution. Removing it after sorting is thus more robust. Overall, **sorting pushes any passage with an anomalously high NPAS toward the end of the retrieved set**, where recomputing NPAS amplifies the poisoning signal due to recency as well, making it easier to detect.
>
> ## 5. How to select $\alpha$ to compute NPAS?
> We introduced $\alpha$ in NPAS specifically to remove any bias in NPAS arising from the length of each passage. Since the idea is to capture all the heavy-hitters in each passage, ideally we should include all the tokens in a passage until there is a sudden drop in attention weight per token.
>
> In practice, selecting $\alpha$ to be larger than the expected token length of the response should be sufficient to capture the heavy-hitters, assuming each token in the response is influenced by at least one token from the relevant passage. Although, if we know that all retrieved passages are of roughly the same length, we can simply consider $\alpha=\infty$.
>
> ## 6. Clean Accuracy on the RQA and NQ datasets.
> As the reviewer mentioned, the clean accuracy of AV Filter is higher than Vanilla RAG by 2% on NQ with Llama-3.1 and 3% on RQA with DeepSeek-R1, while there is no clear reason this occurs specifically on these models or datasets. In general, as observed during the study, the retrieved sets obtained from Google Search are not perfectly aligned with the ground-truth responses in the dataset. There are passages in the retrieved set that are either not fully relevant or provide an incorrect answer (i.e., not consistent with the ground truth). If such passages influence the response too strongly in the benign retrieved set, AV Filter will remove those passages as well, and thus may observe better accuracy than Vanilla RAG. **In an ideal setup where every passage in the benign retrieved set is relevant and aligned with the ground-truth response, this effect should not occur.**
>  \
> \
> **We thank the reviewer for the detailed questions and appreciate the opportunity to provide further insights. We will include each of these clarifications in the revised manuscript to make it more clear and complete. In light of these explanations, we kindly ask the reviewer to consider increasing the rating. We would be happy to provide any further clarifications if needed.**

---

> ### Author Response · Authors · 2025-11-25
> **Request for Reviewer Feedback on Addressed Concerns**
>
> We wanted to follow up and check whether any further clarification is needed from our side. We believe we have addressed the earlier concerns and would appreciate your response when convenient.

---

### Meta-Review · Area_Chair_kc7U · 2026-01-10

**Summary:**

The paper received mixed reviewer scores from 2 to 8. The paper addressed the problem of defending RAG systems against poisoning attacks and was clearly formalized, with proposal to use attention variance as a defense signal and an extensive empirical evaluation across models, datasets, and attack types. However, significant concerns remained regarding the practicality of the assumptions (including access to attention weights and the restriction to sub-majority corruption), the adequacy of comparisons with strong baselines (notably Certified Robust RAG and reranking methods), and the robustness–utility tradeoff, particularly the observed loss in clean accuracy. While the rebuttal clarified several points, it did not sufficiently resolve these core issues. The paper could be strengthened by integrating the rebuttal clarifications into the main text, improving the clarity of the method description.

**Reviewer Concerns:**

The authors clarified the design and limitations of Certified Robust RAG, explaining why its isolate-then-aggregate strategy yields strong ASR but poor utility, especially for multi-hop tasks like HotpotQA, and demonstrated that AV Filter can complement and improve it in all combined settings. Concerns about missing related attacks and defenses (raised mainly by oHcu) were resolved through additional experiments and explicit discussion, leading that reviewer to raise their score from 2 to 6. Questions about reranker baselines were also addressed with justification of the original setup and new results using a stronger commercial reranker, which still showed AV Filter’s advantages. Finally, assumptions about attention access were mitigated by demonstrating effective use of surrogate models for closed-source LLMs.


Multiple reviewers (oHcu, WDZX) noted that the defense fundamentally assumes a minority corruption setting and may fail when poisoned passages dominate retrieval results; while the authors argue this is information-theoretically unavoidable and outline future work on robust retrieval, the limitation still constrains applicability. Similarly, the method is primarily validated on QA-style tasks, leaving its effectiveness on broader generative tasks untested. Reviewer JgSt also remained unconvinced that AV Filter consistently outperforms strong certified baselines in practice, despite clarifications, and noted residual clean accuracy drops that may matter in deployment.

**Reviewer Scores:**

Reviewer WDZX: Likely unchanged at 8 (accept); the rebuttal aligned with expectations and reinforced acknowledged limitations.

Reviewer sWjW: Likely stable at 6, with clarifications resolving technical questions but not fundamentally changing their marginal stance.

Reviewer oHcu: Confirmed increase from 2 to 6 (after Openreview Security Incident), explicitly stating that all concerns were addressed.

Reviewer JgSt: Possibly unchanged (4). The baseline explanations and added reranker results address some concerns, but lingering doubts remain about the necessity of the approach relative to certified baselines.

---

### Decision · Program_Chairs · 2026-01-26

Reject